# Learning to be Smooth:
# An End-to-End Differentiable Particle Smoother

**Ali Younis and Erik B. Sudderth**
Department of Computer Science, University of California, Irvine, CA 92617
ayounis@uci.edu, sudderth@uci.edu

## Abstract

For challenging state estimation problems arising in domains like vision and robotics, particle-based representations attractively enable temporal reasoning about multiple posterior modes. Particle smoothers offer the potential for more accurate offline data analysis by propagating information both forward and backward in time, but have classically required human-engineered dynamics and observation models. Extending recent advances in discriminative training of particle filters, we develop a framework for low-variance propagation of gradients across long time sequences when training particle smoothers. Our "two-filter" smoother integrates particle streams that are propagated forward and backward in time, while incorporating stratification and importance weights in the resampling step to provide low-variance gradient estimates for neural network dynamics and observation models. The resulting mixture density particle smoother is substantially more accurate than state-of-the-art particle filters, as well as search-based baselines, for city-scale global vehicle localization from real-world videos and maps.

## 1  Introduction

Global localization of the state of a moving vehicle using a city-scale map is challenging due to the large area, as well as the inherent ambiguity in urban landscapes, where many street intersections and buildings appear similar. Recent work on global localization [1–11] has typically localized each time point independently during training, sometimes followed by temporal post-processing, often with demanding requirements like near-exact external estimation of relative vehicle poses [11].

For a broader range of state estimation problems in fields like vision and robotics, a number of methods for end-to-end *particle filter* (PF) training have been proposed [8, 12–16]. Learnable PFs are suitable for global localization because they can represent multi-modal posterior densities, propagate uncertainty over time, and learn models of real vehicle dynamics and complex sensors directly from data. However, most learnable PF methods have only been applied to simulated environments [12–14], with only a few preliminary applications to real-world data [8, 15].

Particle filters [17–21] only use past observations to estimate the current state. For offline inference from complete time series, more powerful particle smoothing (PS) algorithms [22–27] may in principle perform better by integrating past and future data. But to our knowledge, recent advances in end-to-end differentiable training of PFs have not been generalized to the more complex PS scenario, requiring error-prone human engineering of PS dynamics and observation models. Classical work on generative parameter estimation via PS [28] is limited to parametric models with few parameters. In contrast, we develop differentiable PS that scale to complex models defined by deep neural networks.

After introducing differentiable particle filters (Sec. 2) and classical particle smoothers (Sec. 3), we develop our differentiable, discriminative *Mixture Density Particle Smoother* (MDPS, see Fig. 1) in Sec. 4. Thorough experiments in Sec. 5 then highlight the advantages of our MDPS over differential PFs on a synthetic bearings-only tracking task, and also show substantial advantages over search-based and retrieval-based baselines for challenging real-world, city-scale global localization problems.

38th Conference on Neural Information Processing Systems (NeurIPS 2024).

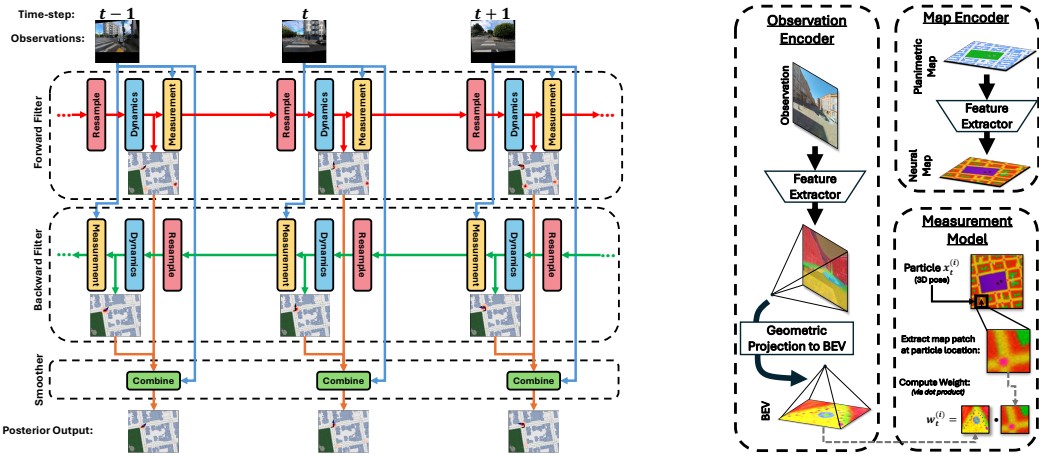

Figure 1: *Left:* Our MDPS method showing the forward and backward particle filters, which are integrated (via learned neural networks, indicated by trapezoids) to produce a smoothed mixture posterior. *Right:* Feature encoders and measurement model used for global localization. First-person camera views are encoded into a Birds-Eye-View (BEV) feature map by extracting features before applying a geometric projection [11]. Map features are extracted via a feed-forward encoder, and un-normalized particle weights are computed as an inner product between BEV features and features of a local map extracted from the global map at the particle location.

## 2 Differentiable Particle Filters

Particle filters iteratively estimate the posterior distribution $p(x_t|y_{1:t}, a_{1:t})$ over the state $x_t$ at discrete time $t$ given a sequence of observations $y_t$ and, optionally, actions $a_t$. PFs use a collection of $N$ samples, or *particles*, $x_t^{(:)} = \{x_t^{(1)}, ..., x_t^{(N)}\}$ with weights $w_t^{(:)} = \{w_t^{(1)}, ..., w_t^{(N)}\}$ to flexibly capture multiple posterior modes nonparametrically. Classic PFs are derived from a Markov generative model, leading to an intuitive recursive algorithm that alternates between proposing new particle locations and updating particle weights. End-to-end training requires gradients for each PF step.

**Particle Proposals.** At each iteration, new particles $x_t^{(:)}$ are proposed by applying a model of the state transition dynamics individually to each particle $x_t^{(i)} \sim p(x_t^{(i)}|x_{t-1}^{(i)}, a_t)$, conditioned on actions $a_t$ if available. Of note, only simulation of the dynamics model is required; explicit density evaluation is unnecessary. Using reparameterization [29–31], the dynamics model can be defined as a feed-forward neural network $f(\cdot)$ that transforms random (Gaussian) noise to produce new particles:

$$x_t^{(i)} = f(\eta_t^{(i)}; x_{t-1}^{(i)}, a_t), \qquad \eta_t^{(i)} \sim N(0, I). \tag{1}$$

**Measurement Updates and Discriminative Training.** Proposed particles are importance-weighted by the likelihood function, $w_t^{(i)} \propto p(y_t|x_t^{(i)})w_{t-1}^{(i)}$, to incorporate the latest observation $y_t$ into the posterior. The updated weights are then normalized such that $\sum_{i=1}^{N} w_t^{(i)} = 1$. However, for complex observations like images or LiDAR, learning accurate generative models $p(y_t|x_t)$ is extremely challenging. Recent work [8, 12–15] has instead learned *discriminative* PFs parameterized by differentiable (typically, deep neural network) *measurement models*:

$$w_t^{(i)} \propto l(x_t^{(i)}; y_t)w_{t-1}^{(i)}. \tag{2}$$

Here, $l(x_t; y_t)$ scores particles to minimize a loss, such as negative-log-likelihood or squared-error, in the prediction of true target states $x_t$ that are observed during training.

### 2.1 Particle Resampling

The stochastic nature of PF dynamics causes some particles to drift towards states with low posterior probability. These low-weight particles do not usefully track the true system state, wasting computational resources and reducing the expressiveness of the overall approximate posterior.

Particle resampling offers a remedy by drawing a new uniformly weighted particle set $\hat{x}_t^{(:)}$ from $x_t^{(:)}$, with each particle duplicated (or not) proportional to its current weight $w_t^{(i)}$. The simplest

*multinomial* resampling strategy [23, 32, 33] chooses particles independently with replacement:

$$\hat{x}_t^{(i)} = x_t^{(j)}, \qquad j \sim \text{Cat}(w_t^{(1)}, \ldots, w_t^{(N)}). \tag{3}$$

To maintain an unbiased posterior, resampled particles have weight $\hat{w}_t^{(i)} = \frac{1}{N}$. Multinomial resampling may be implemented [32] by drawing a continuous $\text{Unif}(0, 1)$ variable for each particle, and transforming these draws by the inverse *cumulative distribution function* (CDF) of particle weights.

*Stratified* resampling [23, 32, 33] reduces the variance of conventional multinomial resampling, by first partitioning the interval $(0, 1]$ into $N$ sub-intervals $(0, \frac{1}{N}] \cup \ldots \cup (1 - \frac{1}{N}, 1]$. One uniform variable is then sampled within each sub-interval, before transforming these draws by the inverse CDF of particle weights. Our differentiable PS incorporate stratified resampling to reduce variance with negligible computational overhead, making training more robustly avoid local optima (see Fig. 2).

While other methods like *residual* resampling [34, 32, 35] have been proposed in the PF literature, this partially-deterministic approach is less robust than stratified resampling in our experiments (see Fig. 2), and also much slower because residual resampling cannot be parallelized across particles.

For our mixture density PS, particles are resampled from a continuous Gaussian mixture, in which all components share a common standard deviation $\beta$. This resampling can equivalently be expressed as $\hat{x}_t^{(i)} = \mu_t^{(i)} + \beta \eta_t^{(i)}$, where $\eta_t^{(i)} \sim N(0, I)$ and $\mu_t^{(i)}$ is generated via discrete sampling of the mixture component means. We incorporate stratified resampling in this step to boost performance.

## 2.2 Differentiable Approximations of Discrete Resampling

For discriminative PFs to effectively learn to propagate state estimates over time, gradients are needed for all steps of the PF. While differentiable dynamics and measurement models are easily constructed via standard neural-network architectures, discrete particle resampling is *not* differentiable.

**Truncated-Gradient Particle Filters (TG-PF)** [15], the first so-called "differentiable" particle filter, actually treated the resampling step as non-differentiable and simply truncated gradients to zero at resampling, preventing *back-propagation through time* (BPTT) [36]. Due to this weakness, dynamics models were assumed known rather than learned, and measurement models were learned from biased gradients that fail to propagate information over time, reducing accuracy [14].

**Soft Resampling Particle Filters (SR-PF)** [13] utilize a differentiable resampling procedure that sets particle resampling weights to be a mixture of the true weights and a discrete uniform distribution:

$$\hat{x}_t^{(i)} = x_t^{(j)}, \qquad j \sim \text{Cat}(v_t^{(1)}, \ldots, v_t^{(N)}), \qquad v_t^{(i)} = (1 - \lambda) w_t^{(i)} + \frac{\lambda}{N}. \tag{4}$$

Gradients are then propagated via the resampled particle weights defined as:

$$\hat{w}_t^{(i)} = \frac{w_t^{(i)}}{(1 - \lambda) w_t^{(i)} + \lambda/N}, \qquad \nabla_\phi \hat{w}_t^{(i)} = \nabla_\phi \left( \frac{w_t^{(i)}}{(1 - \lambda) w_t^{(i)} + \lambda/N} \right). \tag{5}$$

This simple approach resamples low-weight particles more frequently, degrading performance. The gradients of Eq. (5) also have substantial bias, because they incorrectly assume model perturbations only influence the particle weights in (5), and not the discrete particle resampling in (4).

**Relaxations of Discrete Resampling.** While discrete particle resampling could potentially be replaced by continuous particle interpolation with samples from a Gumbel-Softmax or Concrete distribution [37, 38], no work has successfully applied such relaxations to PFs, and experiments in Younis and Sudderth [14] show very poor performance for this baseline. Alternatively, *entropy-regularized optimal transport particle filters* (OT-PF) [12] replace discrete resampling with an entropy-regularized optimal transport problem, that minimizes a Wasserstein metric to determine a probabilistic mapping between the weighted pre-resampling particles and uniformly weighted post-resampling particle. OT-PF performance is sensitive to a non-learned entropy regularization hyperparameter, and the biased gradients induced by this regularization may substantially reduce performance [14]. Furthermore, "fast" Sinkhorn algorithms [39] for entropy-regularized OT still scale quadratically with the number of particles, and in practice are orders of magnitude slower than competing resampling strategies. This makes OT-PF training prohibitively slow on the challenging city-scale localization tasks considered in this paper, so we do not compare to it.

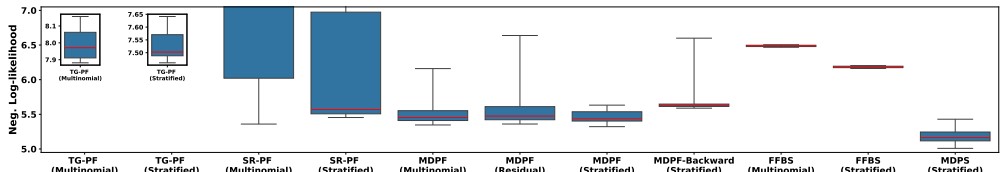

Figure 2: Box plots showing median (red line), quartiles (blue box), and range (whiskers) over 11 training runs for Bearings-Only tracking (Sec. 5.1). We boost the robustness of the top-performing MDPF [14], which previously used multinomial resampling, by incorporating variance-reduced stratified resampling; residual resampling is both slower and less effective. Stratified resampling provides larger advantages for the less-sophisticated TG-PF [15] and SR-PF [13] gradient estimators, but these baselines remain inferior to MDPF. Our MDPS substantially improves on all PFs by incorporating both past and future observations when computing posteriors. Classic FFBS particle smoothers [24, 25] have poor performance, even when provided the true likelihoods (rather than a learned approximation), showing the effectiveness of our end-to-end learning of particle proposals and weights. Forward PFs are initialized with noisy samples of the true state, while MDPF-Backward (the backwards-time PF component of MDPS) is initialized by sampling uniformly from the state space.

## 2.3 Mixture Density Particle Filters

*Mixture Density Particle Filters* (MDPF) [14] are a differentiable variant of regularized PFs [40, 41]. MDPF estimates a continuous *kernel state density* [42] by convolving particles with a continuous, and differentiable, kernel function $K$ (such as a Gaussian) with bandwidth hyperparameter $\beta$:

$$m(x_t \mid x_t^{(:)}, w_t^{(:)}, \beta) = \sum_{i=1}^{N} w_t^{(i)} K(x_t - x_t^{(i)}; \beta). \tag{6}$$

Particles are then resampled $\hat{x}_t^{(i)} \sim m(x_t \mid x_t^{(:)}, w_t^{(:)}, \beta)$ from this continuous mixture instead of via discrete resampling. Unbiased and low-variance *Importance Weighted Sample Gradient* (IWSG) [14] estimates may then be constructed by viewing the particle proposal $q(z) = m(z \mid \phi_0)$ to be fixed to the mixture model parameters $\phi_0 = \{x_t^{(:)}, w_t^{(:)}, \beta\}$ at the current training iteration. Gradients then account for parameter perturbations *not* by perturbing particle locations as in standard reparameterization [29–31], but by perturbing particle importance weights away from uniform:

$$\hat{w}^{(i)} = \frac{m(z^{(i)} \mid \phi)\big|_{\phi=\phi_0}}{m(z^{(i)} \mid \phi_0)} = 1, \qquad \nabla_\phi \hat{w}^{(i)} = \frac{\nabla_\phi m(z^{(i)} \mid \phi)\big|_{\phi=\phi_0}}{m(z^{(i)} \mid \phi_0)}. \tag{7}$$

With this approach, the bandwidth parameter $\beta$ may also be tuned for end-to-end prediction of state distributions, avoiding the need for classic bandwidth-selection heuristics [42–44]. An "adaptive" variant of MDPF [14] incorporates two bandwidths, one for particle resampling (to propagate information over time) and a second for estimation of state posteriors (to compute the loss). Our MDPS also incorporates separate bandwidths for resampling and state estimation, as detailed below.

## 3 From Filtering to Smoothing

Particle smoothers extend PFs to estimate the state posteriors $p(x_t|y_{1:T})$ given a full $T$-step sequence of observations. (To simplify equations, we do not explicitly condition on actions $a_{1:T}$ in the following two sections.) Particle smoothers continue to approximate posteriors via a collection of particles $\overleftrightarrow{x}_t^{(1:N)}$ with associated weights $\overleftrightarrow{w}_t^{(1:N)}$, where we use bi-directional overhead arrows to denote smoothed posteriors. Classical particle smoothing algorithms, which are non-differentiable and typically assume human-engineered dynamics and likelihoods, fall into two broad categories.

**Forward-Filtering, Backward Smoothing (FFBS)** algorithms [24, 25] compute $p(x_t|y_{1:T})$ by factoring into forward filtering and backward smoothing components:

$$p(x_t|y_{1:T}) = \int p(x_t, x_{t+1}|y_{1:T})dx_{t+1} = \underbrace{p(x_t|y_{1:t})}_{\text{forward filtering}} \underbrace{\int \frac{p(x_{t+1}|y_{1:T})p(x_{t+1}|x_t)}{\int p(x_{t+1}|x_t)p(x_t|y_{1:t})}dx_{t+1}}_{\text{backward smoothing}}. \tag{8}$$

A natural algorithm emerges from Eq. (8), where a conventional PF first approximates $p(x_t|y_{1:t})$ for all times via particles $\overrightarrow{x}_t^{(1:N)}$ with weights $\overrightarrow{w}_t^{(1:N)}$. A backward smoother then recursively reweights

the "forward" particles to account for future data, but does *not* change particle locations:

$$\overleftrightarrow{w}_t^{(i)} \propto \overrightarrow{w}_t^{(i)} \left( \sum_{j=1}^{N} \overleftrightarrow{w}_{t+1}^{(j)} \frac{p(\overrightarrow{x}_{t+1}^{(j)}|\overrightarrow{x}_t^{(i)})}{\sum_{k=1}^{N} \overrightarrow{w}_t^{(k)} p(\overrightarrow{x}_{t+1}^{(j)}|\overrightarrow{x}_t^{(k)})} \right). \tag{9}$$

Because FFBS sets $\overleftrightarrow{x}_t^{(i)} = \overrightarrow{x}_t^{(i)}$, it is only effective when filtered state posteriors $p(x_t|y_{1:t})$ substantially overlap with smoothed posteriors $p(x_t|y_{1:T})$ [25]; performance deteriorates when future data is highly informative. FFBS also requires explicit evaluation, not just simulation, of the state transition dynamics $p(x_{t+1}|x_t)$, which is not tractable for dynamics parameterized as in Eq. (1).

**Two Filter Smoothing (TFS)** algorithms [22, 24, 25] instead express the smoothed posterior as a normalized product of distinct forward-time and backward-time filters:

$$p(x_t|y_{1:T}) = \frac{p(x_t|y_{1:t})p(y_{t+1:T}|x_t)}{p(y_{t+1:T}|y_{1:t})} \propto p(x_t|y_{1:t})p(y_{t+1:T}|x_t). \tag{10}$$

Here $p(x_t|y_{1:t})$ may be approximated by a standard PF, and $p(y_{t+1:T}|x_t)$ is the so-called *backward information filter* [25, 24] defined as

$$p(y_{t:T}|x_t) = \int p(y_{t+1:T}|x_{t+1})p(x_{t+1}|x_t)p(y_t|x_t)dx_{t+1}. \tag{11}$$

Because $p(y_{t:T}|x_t)$ is a likelihood function rather than a probability density in $x_t$, and it is possible that $\int p(y_{t:T}|x_t)dx_t = \infty$. This is not an issue when $p(y_{t:T}|x_t)$ is computed analytically as in Kalman smoothers for Gaussian models [45], but particle-based methods can only hope to approximate finite measures. Bresler [22] addresses this issue via an *auxiliary* probability measure $\gamma_t(x_t)$:

$$p(y_{t:T}|x_t) \propto \frac{\tilde{p}(x_t|y_{t:T})}{\gamma_t(x_t)}, \qquad \tilde{p}(x_{t:T}|y_{t:T}) \propto \gamma_t(x_t)p(y_t|x_t) \prod_{s=t+1}^{T} p(x_s|x_{s-1})p(y_s|x_s). \tag{12}$$

From Eqs. (10,12), the smoothed posterior is a reweighted product of forward and backward filters:

$$p(x_t|y_{1:T}) \propto \frac{\overbrace{p(x_t|y_{1:t})}^{\text{forward filtering}} \overbrace{\tilde{p}(x_t|y_{t+1:T})}^{\text{backward filtering}}}{\gamma_t(x_t)}. \tag{13}$$

This suggest an algorithm where two PFs are run on the sequence independently, one forward and one backward in time, to compute forward particles $\{\overrightarrow{x}_t^{(1:N)}, \overrightarrow{w}_t^{(1:N)}\}$ and backward particles $\{\overleftarrow{x}_t^{(1:N)}, \overleftarrow{w}_t^{(1:N)}\}$. Because continuously sampled forward and backward particle sets will not exactly align, classic TFS integrate these two filters by rewriting Eq. (13) as follows:

$$p(x_t|y_{1:T}) \propto \frac{p(y_t|x_t)\tilde{p}(x_t|y_{t+1:T}) \int p(x_t|x_{t-1})p(x_{t-1}|y_{1:t-1})dx_{t-1}}{\gamma_t(x_t)}. \tag{14}$$

This yields a particle re-weighting approach where backward filter particles $\overleftarrow{x}_t^{(1:N)}$ are re-weighted using the forward filter particle set, to produce the final smoothed particle weights $\overleftrightarrow{w}_t^{(1:N)}$:

$$\overleftrightarrow{w}_t^{(i)} \propto \overleftarrow{w}_t^{(i)} \sum_{j=1}^{N} \overrightarrow{w}_{t-1}^{(j)} \frac{p(\overleftarrow{x}_t^{(i)}|\overrightarrow{x}_{t-1}^{(j)})}{\gamma_t(\overleftarrow{x}_t^{(i)})}, \qquad \sum_{i=1}^{N} \overleftrightarrow{w}_t^{(i)} = 1. \tag{15}$$

Conventional TFS set $\overleftrightarrow{x}_t^{(1:N)} = \overleftarrow{x}_t^{(1:N)}$, which similar to FFBS, makes performance heavily dependant on significant overlap in support between $p(x_t|y_{t+1:T})$ and $p(x_t|y_{1:T})$. Like FFBS, TFS also restrictively requires evaluation (not just simulation) of the state transition dynamics.

## 4 Mixture Density Particle Smoothers

Our novel *Mixture Density Particle Smoother* (MDPS, Fig. 1) can be seen as a differentiable TFS, where the forward and backward filters of Eq. (13) are defined as MDPFs (Sec. 2.3). Using discriminative differentiable particle filters (MDPFs) within the TFS frameworks, and replacing Eq. (14) with an importance-weighted integration of forward and backward particles, enables an effective and end-to-end differentiable particle smoother. We begin by rewriting Eq. (13) as

$$p(x_t|y_{1:T}) \propto \frac{p(y_t|x_t)p(x_t|y_{1:t-1})\tilde{p}(x_t|y_{t+1:T})}{\gamma_t(x_t)}, \tag{16}$$

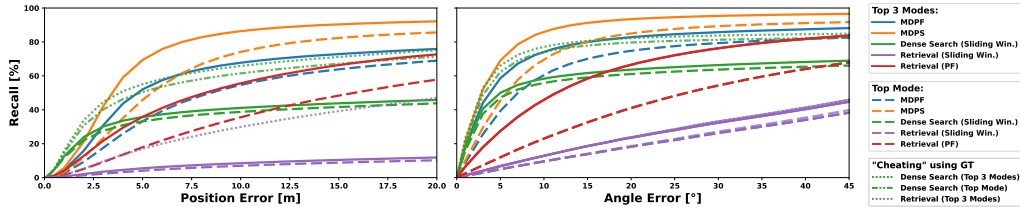

Figure 3: Position and error recall using the MGL [11] dataset. Recall is computed with the top posterior mode as well as with the best of the top-3 posterior modes, extracted via non-maximal suppression. As expected, Retrieval [3] methods do poorly due to their lack of discrimination power between neighboring map patches. Dense search [11] does better by using fine map details during localization, but it requires a ground truth hint ("Cheating" with GT, which artificially improves performance) to work well at city-scale environments. Retrieval (PF) [8] uses unlearned state dynamics, which proves useful, but still suffers from the poor discriminative ability of retrieval. In contrast, MDPF [14] uses end-to-end learned dynamics and measurement models, allowing for good performance but suffering from only using past information when estimating posterior densities. Our MDPS is able to learn similar strong dynamics and measurement models as MDPF, and also incorporates future as well as past information to achieve a more accurate posterior density and thus higher recall.

where the forward and backward filters no longer condition on the current observation. This allows for functionally identical MDPFs to be used for both directions, simplifying implementation. MDPFs parameterize state posteriors as continuous kernel density mixtures:

$$p(x_t|y_{1:t-1}) = \sum_{i=1}^{N} \overrightarrow{w}_t^{(i)} K(x_t - \overrightarrow{x}_t^{(i)}; \overrightarrow{\beta}), \qquad p(x_t|y_{t+1:T}) = \sum_{i=1}^{N} \overleftarrow{w}_t^{(i)} K(x_t - \overleftarrow{x}_t^{(i)}; \overleftarrow{\beta}). \quad (17)$$

Unlike discrete probability measures, these continuous mixture distributions can be combined via direct multiplication to give a smoothed posterior mixture containing $N^2$ components, one for each pair of forward and backward particles. For this product integration of forward and backward filters, the normalizing constants for all pairs of kernels must be explicitly computed to correctly account for the degree to which hypotheses represented by forward and backward particles are consistent. These normalization constants are tractable for some simple kernels including Gaussians [46], but more complex for the other cases such as von Mises kernels of orientation angles [47, 48].

Direct mixture multiplication eliminates the need to evaluate the dynamics model, as in classic TFS, but introduces significant overhead due to the quadratic scaling of the number of mixture components. To address this issue, our MDPS uses importance sampling where the smoothed posterior is defined by $M \ll N^2$ particles drawn from a mixture of the filter posteriors:

$$\overleftrightarrow{x}_t^{(i)} \sim q(x_t) = \frac{1}{2}p(x_t|y_{1:t-1}) + \frac{1}{2}p(x_t|y_{t+1:T}), \qquad i = 1, \dots, M. \quad (18)$$

By construction, this proposal will include regions of the state space that lie within the support of *either* $p(x_t|y_{1:t-1})$ or $p(x_t|y_{t+1:T})$, improving robustness. Our experiments set $M = 2N$, drawing $N$ particles from each of the filtered and smoothed posteriors. Given true dynamics and likelihood models, importance sampling may correct for the fact that smoothed particles are drawn from a mixture rather than a product of filtered densities, as well as incorporate the local observation:

$$\overleftrightarrow{w}_t^{(i)} \propto \frac{p(y_t|\overleftrightarrow{x}_t^{(i)})p(\overleftrightarrow{x}_t^{(i)}|y_{1:t-1})\tilde{p}(\overleftrightarrow{x}_t^{(i)}|y_{t+1:T})}{\gamma_t(\overleftrightarrow{x}_t^{(i)})q(\overleftrightarrow{x}_t^{(i)})}, \qquad \sum_{i=1}^{M} \overleftrightarrow{w}_t^{(i)} = 1. \quad (19)$$

To more easily train a discriminative PS, rather than estimating each term in Eq. (19) separately, we directly parameterize their product via a feed-forward neural network $l(\cdot)$:

$$\overleftrightarrow{w}_t^{(i)} \propto \frac{l(\overleftrightarrow{x}_t^{(i)}; y_t, p(\overleftrightarrow{x}_t^{(i)}|y_{1:t-1}), \tilde{p}(\overleftrightarrow{x}_t^{(i)}|y_{t+1:T}))}{q(\overleftrightarrow{x}_t^{(i)})}, \qquad \sum_{i=1}^{M} \overleftrightarrow{w}_t^{(i)} = 1. \quad (20)$$

The posterior weight network $l(\cdot)$ scores particles based on agreement with $y_t$, as well as consistency with the forward and backward filters, and implicitly accounts for the auxiliary distribution $\gamma_t(\cdot)$. To allow state prediction and compute the training loss, the smoothed posterior is approximated as:

$$p(x_t|y_{1:T}) \approx m(x_t|\overleftrightarrow{x}_t^{(:)}, \overleftrightarrow{w}_t^{(:)}, \overleftrightarrow{\beta}) = \sum_{i=1}^{N} \overleftrightarrow{w}_t^{(i)} K(x_t - \overleftrightarrow{x}_t^{(i)}; \overleftrightarrow{\beta}), \quad (21)$$

where $\overleftrightarrow{\beta}$ is a learned, dimension-specific bandwidth parameter.

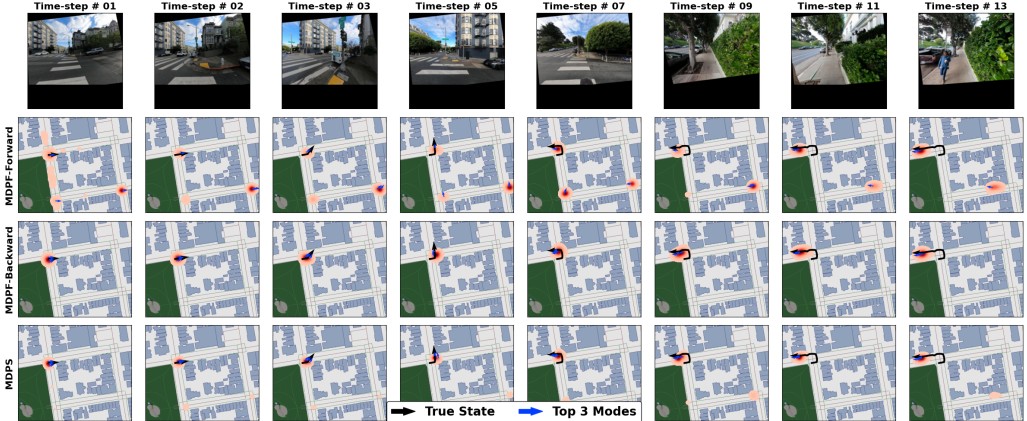

Figure 4: Example trajectories from the MGL dataset with observations shown in the top row. We show the current true state and state history (black arrow and black line), the estimated posterior density of the current state (red cloud, with darker being higher probability) and the top 3 extracted modes (blue arrows) for the MDPS as well as its forward and backward MDPFs. Due to ambiguity at early time-steps, MDPF [14] is unable to resolve the correct intersection, and instead places probability mass at multiple intersections. By fusing both forward and backward filters, our MDPS resolves this ambiguity with probability mass focused on the correct intersection. Furthermore, MDPS provides a tighter posterior density than either MDPF-Forward or MDPF-Backward.

**Training Loss and Gradient Computation**. We discriminatively train our MDPS by minimizing the negative log-likelihood of the true state sequence:

$$\mathcal{L} = \frac{1}{T} \sum_{t \in T} -\log(m(x_t | \overleftrightarrow{x}_t^{(:)}, \overleftrightarrow{w}_t^{(:)}, \overleftrightarrow{\beta})). \tag{22}$$

During training, the IWSG estimator of Eq. (7) provides unbiased estimates of the gradients of the forward and backward resampling steps. We may similarly estimate gradients of the mixture resampling (18) that produces smoothed particles, enabling the first end-to-end differentiable PS:

$$\nabla_\phi \overleftrightarrow{w}_t^{(i)} \propto \frac{\nabla_\phi l(\overleftrightarrow{x}_t^{(i)}; y_t, p(\overleftrightarrow{x}_t^{(i)} | y_{1:t-1}), \tilde{p}(\overleftrightarrow{x}_t^{(i)} | y_{t+1:T}))}{q(\overleftrightarrow{x}_t^{(i)})}. \tag{23}$$

**Training Details.** Because the smoother weights of Eq. (20) cannot be effectively learned when filter parameters are random, we train MDPS via a three-stage procedure. In stage 1, the forward and backward PFs are trained separately (sharing only parameters for the encoders, see Fig. 1) to individually predict the state. In stage 2, the PFs are frozen and the particle smoother measurement model $l(\cdot)$ of Eq. (20) is trained. In stage 3, all models are unfrozen and trained jointly to minimize the loss in the MDPS output state posterior predictions of the true states. The forward MDPF, backward MDPF, and MDPS posterior each have separate kernel bandwidths ($\overrightarrow{\beta}$, $\overleftarrow{\beta}$, $\overleftrightarrow{\beta}$) that are jointly learned with the dynamics and measurement models. We randomly resample a stochastic subset of the training sequences for each step, and adapt learning rates via the Adam [49] optimizer.

**Computational Requirements.** At training time, to allow unbiased gradient propagation, MDPS computes importance weights for each particle during resampling. For $N$ particles and $T$ time-steps, this requires $\mathcal{O}(TN^2)$ operations. At inference time, importance weighting is not needed as the particle weights can simply be set as uniform, and resampling only requires $\mathcal{O}(TN)$ operations. All phases of our MDPS scale linearly with $N$ at test time, in contrast with other differentiable relaxations such as OT-PF [12], which requires $\mathcal{O}(TN^2)$ operations for both training and inference.

## 5  Experiments

We evaluate our MDPS on a synthetic bearings-only tracking task [14], as well as on real-world city-scale global localization. For all tasks, we estimate the MDPF/MDPS posterior distributions of a 3D (translation and angle) state $x_t = (x, y, \theta)$, using Gaussian kernels for the position dimensions, and von Mises kernels for the angular dimensions of the state posterior mixtures.

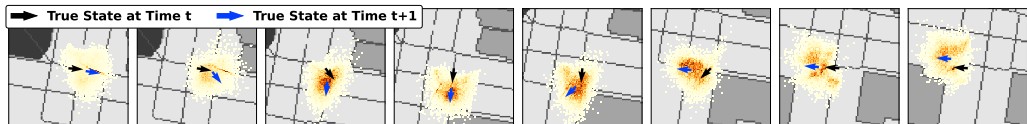

Figure 5: Learned dynamics from the forward filter of MDPS trained on the MGL dataset. Density cloud illustrates density of particles after applying dynamics while marginalizing actions. MDPS clearly learns informative, non-linear dynamics models which aid in state posterior estimation.

## 5.1 Bearings Only Tracking Task

To allow comparison to prior discriminative PFs, we use the same bearings-only tracking task as [14], where the 3D state of a variable-velocity synthetic vehicle is tracked via noisy bearings from a fixed-position radar. 85% of observations are the true bearing plus von Mises noise, while 15% are uniform outliers. Train and evaluation sequences have length $T = 50$. Unlike Younis and Sudderth [14], we find truncated BPTT [50] is not necessary if bandwidths are initialized appropriately. Filtering particles are initialized as the true state with added Gaussian noise, while MDPF-Backward (and the MDPS backwards filter) are initialized with uniformly sampled particles to mimic datasets where often only the starting state is known. More details can be found in the Appendix.

We compare our MDPS methods to several existing differentiable particle filter baselines, but no differentiable particle smoother baseline exists. Instead, we implement the classic FFBS [24, 25] algorithm (Sec. 3), which assumes known dynamics and measurement models. Since FFBS is not differentiable, we learn the dynamics model using the dataset true state pairs $\{x_{t-1}, x_t\}$ outside of the FFBS algorithm. In order to simulate from and evaluate the state transition dynamics, as needed by the FFBS, we parameterize the dynamics model to output a mean and use a fixed bandwidth parameter (tuned on validation data) to propose new particles. We also use the true observation likelihood as the measurement model, instead of a learned approximation; this boosts FFBS performance.

**Results.** In Fig. 2 we show statistics of performance over 11 training and evaluation runs for each method. We compare to TG-PF [15], SR-PF [13], the classical FFBS [24, 25], and MDPF [14]. Interestingly, MDPF outperforms SR-PF and TG-PF even when the initial particle set is drawn uniformly from the state space as in MDPF-Backward.

By incorporating more temporal data, MDPS substantially outperforms MDPF. Even when unfairly provided the true observation likelihood, FFBS performs poorly since particles are simply re-weighted (not moved) by the backward smoother. This inflexibility, and lack of end-to-end learning, makes FFBS less robust to inaccuracies in the forward particle filter.

We are the first to compare resampling variants in the context of modern discriminative PFs. Stratified resampling substantially improves TG-PF and SR-PF performance, but only modestly improves the worst-performing MDPF runs. This may be because even with basic multinomial resampling, the lower-variance MDPF gradients dramatically outperform all TG-PF and SR-PF variants. Residual resampling performs worse than stratified resampling, and is also much slower since it cannot be easily parallelized on GPUs, so we do not consider it for other datasets.

## 5.2 City Scale Global Localization Task

Our global localization task is adopted from Sarlin et al. [11], where we wish to estimate the 3D state (position and heading) of a subject (person/bike/vehicle) as it moves through real-world city-scale environments. Observations are gravity-aligned first-person images, actions are noisy odometry, and a 2D planimetric map is provided to help localize globally. We use the Mapillary Geo-Localization [11] and KITTI [51] datasets to compare our MDPS method to MDPF [14] as well as other methods specifically designed for the global localization task, which are not sequential Monte Carlo methods.

Our global localization task is distinct from local localization systems, which aim to track subject positions relative to their starting position, instead of in relation to the global map origin. Visual SLAM systems [21] almost exclusively solve the local localization task, using the starting position as the origin of their estimated map. If a map is provided, then just the localization part of Visual SLAM can be run, but detailed visual or 3D maps of the environment are needed. These have prohibitive memory requirements at city-scales, and need constant updating as the visual appearance of the environment changes (e.g., with the weather/seasons) [11]. Hybrid place recognition with localization

also requires detailed visual or 3D maps [52]. In our experiments, we instead seek to use planimetric maps for global localization, which are compact and robust to environment changes. It is not obvious how to apply SLAM/Hybrid place recognition systems to this type of map.

**Retrieval methods** [1–7] rely on latent vector similarity where map patches and the observation are encoded into a common latent state space before doing a vector similarity search. These methods are trained using a contrastive loss [2, 53] that forces the observation encoding to be similar to its corresponding map patch encoding, while being dissimilar to other patch and observation encodings. Accuracy depends on map patch extraction density and patch similarity; if similar patches are mapped to near-identical encodings, performance suffers. Zhou et al. [8] extend this framework using a non-differentiable PF, where a retrieval-based measurement model is trained outside the PF framework, and non-learned dynamics are fixed to actions with added Gaussian noise.

**Refinement methods** [9, 10] refine an initial position estimate via expensive optimization, by maximizing the alignment of features extracted from the observation and map. Due to the extreme non-linearity of this objective, refinement methods require an accurate initial estimate to converge to the correct solution, if at all. This prevents their use for city-scale global localization.

**Dense Search** [11] extracts *birds-eye-view* (BEV) features from the observed images via geometric projection (see Fig. 1), before applying a dense search to align BEV features with extracted map features. Heuristic alignment probabilities may be produced by tracking alignment values during search, and applying a softmax operator. Higher discretization density boosts accuracy, but requires significantly more memory and compute. Temporal information can be used by warping probability volumes onto the current time-step, but this requires near-exact relative poses which Sarlin et al. [11] determine via an expensive, black-box Visual-Inertial SLAM system [21].

## 5.3 Mapillary Geo-Localization (MGL) Dataset

In the Mapillary Geo-Localization (MGL) [11] dataset, images sequences are captured from handheld or vehicle-mounted cameras as a subject (person/bike/car) roams around various European and U.S. cities. Observations are set as $90°$ Field-of-View images in various viewing directions, with actions being noisy odometry. A planimetric map of the environment is also provided via the OpenStreetMap platform [54] at 0.5 meter/pixel resolution. We generate custom training, validation, and test splits to create longer sequences with $T = 100$ steps. For particle-based methods, we use stratified resampling and set the initial particle sets to be the true state with added noise. More details are in the Appendix.

**Implementation Details.** For MDPF and our MDPS, we set the dynamics model to a multi-layer perceptron (MLP) network. The measurement model incorporates BEV features [11] and map features as illustrated in Fig. 1. The smoother measurement model incorporates additional inputs via an extra MLP. Memory constraints prevent Dense Search in city-scale environments, so we consider two methods to limit the search space. A sliding window limits the search space to a $256m \times 256m$ area that is recursively re-centered around the best position estimate at $t - 1$, propagated to $t$ using $a_t$. We can also limit the search space to a $256m \times 256m$ area containing the true state, though this *artificially* increases performance. We similarly limit the search space for Retrieval, which performs poorly in large environments; Zhou et al. [8] even limit the vector search to known road networks.

**Results.** We compare our MDPS to MDPF [14], Dense Search [11], Retrieval [3] implemented as detailed by [11], and Retrieval (PF) [8], reporting the position and rotation recall in Fig. 3. Due to ambiguity in large city environments (e.g., intersections can look very similar), estimated state posteriors can be multi-modal (see Fig. 4), and thus simply reporting accuracy using the highest probability mode does not fully characterize performance. We thus also extract the top-three modes using non-maximal suppression (see Appendix), and report accuracy of the best mode. Interestingly, MDPF and MDPS give dramatic improvements over baselines engineered specifically for this task, highlighting the usefulness of end-to-end training. MDPS outperforms MDPF by using the full sequence of data to reduce mode variance, and discard incorrect modes as illustrated in Fig. 4.

Informative dynamics models boost performance, as demonstrated by the MDPS and MDPF results in Fig. 3. We visualize the learned dynamics of the MDPS forward filter in Fig. 5. Good dynamics models keep particles densely concentrated in high-probability regions, while also including diversity to account for sometimes-noisy actions. This enables learning of more discriminative measurement models, since training encourages the weights model to disambiguate nearby particles.

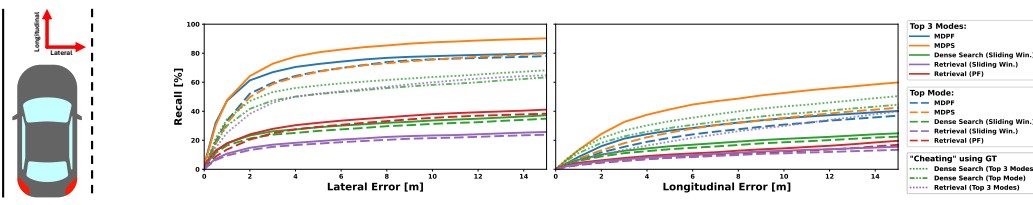

Figure 6: *Left:* Lateral and longitudinal errors are in the vehicle frame of reference. *Right:* Position recall versus error for the KITTI [51] dataset. Recall is computed with the top posterior mode as well as with the best of the top-3 posterior modes. Longitudinal localization performance is poor for all methods due to lack of visual features. MDPF [14] offers dramatic improvements for lateral error over Retrieval [3], Retrieval (PF) [8] and Dense Search [11] baselines, even when these baselines are constrained to operate around the ground truth state ("Cheating" with GT). For longitudinal recall, methods using "Cheating" with GT have good performance because they are *artificially* constrained to be near the true state, and thus have significantly less position ambiguity along the roadway. MDPS offers further improvements over MDPF as it maintains a more diverse set of posterior modes, instead of prematurely collapsing to incorrect modes.

**Computational Requirements.** While MDPS is more accurate than other methods, it is also more efficient than dense search. Because dense search must try many options to find the best alignment of the BEV and extracted map features, it has complexity $\mathcal{O}(KW^2H^2)$ for $K$ search rotations, and search locations defined on a grid of width $W$ and height $H$. (For notational simplicity, we assume the BEV features and the map features have the same width and height.) This complexity can be reduced to $\mathcal{O}(KWH\log(WH))$ by using the Fast Fourier Transform. In contrast, MDPS has complexity $\mathcal{O}(NWH)$, where $N \ll K\log(WH)$, as it only compares the BEV and map features at the particle locations. This allows MDPS to better scale to large-scale environments.

### 5.4 KITTI Dataset

We also evaluate our MDPS method for the global localization task using the KITTI [51] dataset, where observations are forward-facing camera images from a moving vehicle. We augment this datatset with noisy odometry computed from the ground truth states and use the default *Train*, *Test1*, and *Test2* splits for training, validation, and evaluation respectively. Like MGL, a planimetric map of the environment is provided via the OpenStreetMap platform [54] at 0.5 meter/pixel resolution. Due to the small size of the KITTI dataset, we pre-train all methods using MGL before refining on KITTI, using the same network architectures as was used for the MGL dataset. See Appendix for details.

**Results.** Due to the forward-facing camera, the observation images lack visual features for useful localizing along the roadway, therefore we decouple the position error into lateral and longitudinal errors when reporting recalls in Fig. 6. Understandably, all methods have larger longitudinal error than lateral error. Interestingly, MDPF and MDPS offer similar Top 3 mode performance for small lateral errors (under 2 meters) while significantly outperforming all other methods. When the lateral error is greater than 2 meters, MDPS sees a performance gain as it maintains a more diverse set of posterior modes, whereas MDPF prematurely collapses the posterior density to incorrect modes.

### 5.5 Limitations

Like all particle-based methods, our MDPS suffers from the *curse of dimensionality* [55] where particle sparsity increases as the dimension of the state space increases, reducing expressiveness of state posteriors. More effective use of particles via smarter dynamics and measurement models, as enabled by end-to-end MDPS training, can reduce but not eliminate these challenges.

## 6   Discussion

We have developed a fully-differentiable, two-filter particle smoother (MDPS) that robustly learns more accurate models than classic particle smoothers, whose components are often determined via heuristics. MDPS successfully incorporates temporal information from the past as well as the future to produce more accurate state posteriors than state-of-the-art differentiable particle filters. When applied to city-scale global localization from real imagery, our learned MDPS dramatically improves on search and retrieval-based baselines that were specifically engineered for the localization task.

## Acknowledgments

This research supported in part by NSF Robust Intelligence Award No. IIS-1816365 and ONR Award No. N00014-23-1-2712.

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

# A Additional Experiment Results

In this section we give additional experiment results for the global localization task on the MGL [11] and KITTI [51] datasets.

## A.1 MGL Dataset Additional Results

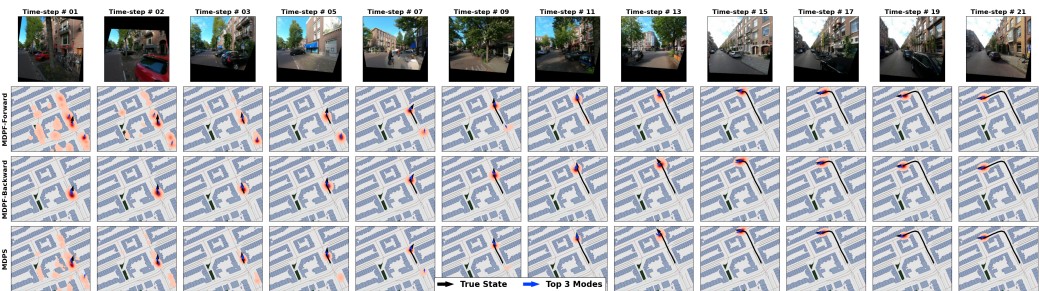

Figure 7: Additional example trajectories from the MGL dataset with observations shown in the top row. We show the current true state and state history (black arrow and black line), the estimated posterior density of the current state (red cloud, with darker being higher probability) and the the top 3 extracted modes (blue arrows) for each method. By using the full sequence of observations and actions when computing state posteriors, MDPS is able to estimate a more accurate and tighter posterior than the forward or backward MDPFs.

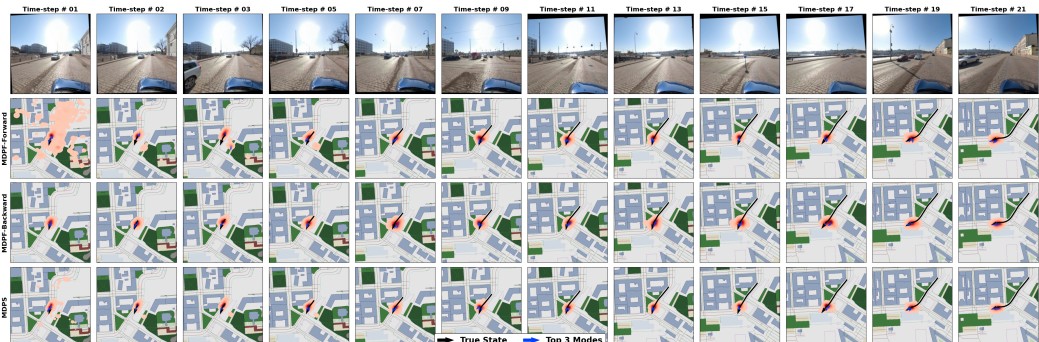

Figure 8: Additional example trajectories from the MGL dataset with observations shown in the top row. We show the current true state and state history (black arrow and black line), the estimated posterior density of the current state (red cloud, with darker being higher probability) and the the top 3 extracted modes (blue arrows) for each method. By using the full sequence of observations and actions when computing state posteriors, MDPS is able to estimate a more accurate and tighter posterior than the forward or backward MDPFs.

## A.2 Kitti Dataset Additional Results

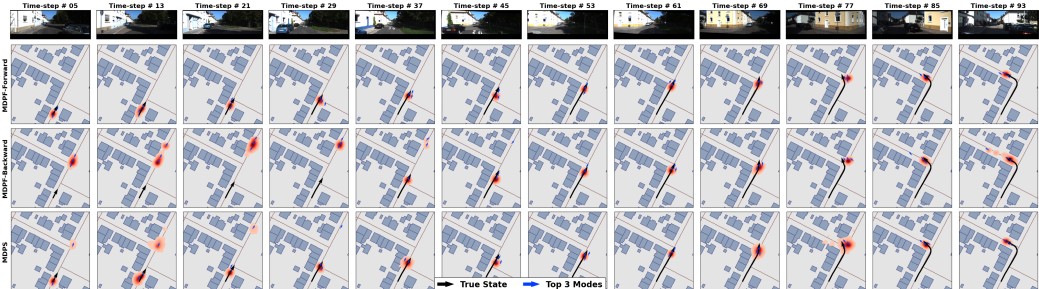

Figure 9: Additional example trajectories from the KITTI dataset with observations shown in the top row. We show the current true state and state history (black arrow and black line), the estimated posterior density of the current state (red cloud, with darker being higher probability) and the the top 3 extracted modes (blue arrows) for each method.

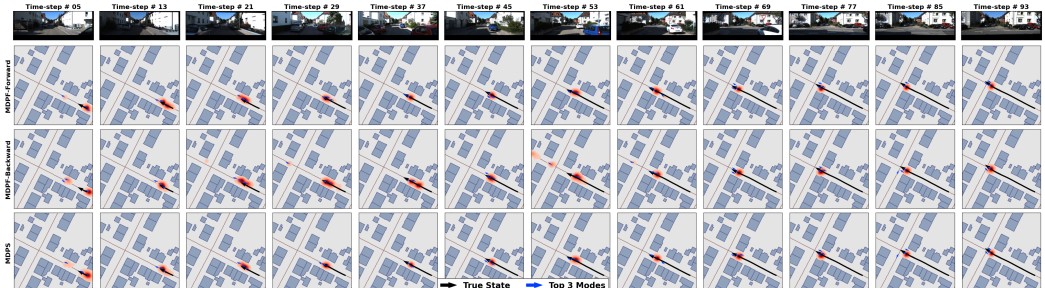

Figure 10: Additional example trajectories from the KITTI dataset with observations shown in the top row. We show the current true state and state history (black arrow and black line), the estimated posterior density of the current state (red cloud, with darker being higher probability) and the the top 3 extracted modes (blue arrows) for each method.

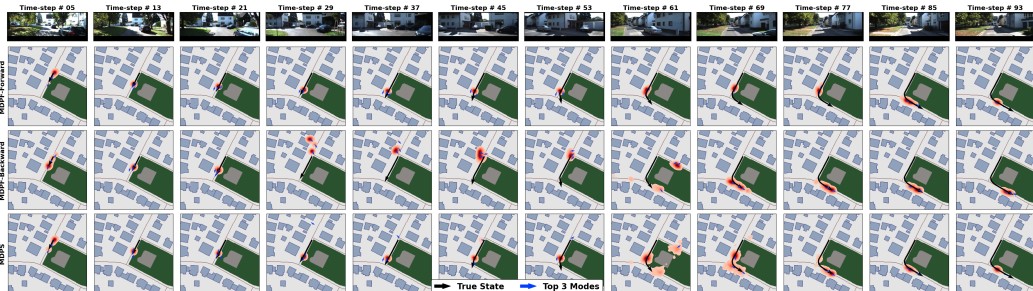

Figure 11: Additional example trajectories from the KITTI dataset with observations shown in the top row. We show the current true state and state history (black arrow and black line), the estimated posterior density of the current state (red cloud, with darker being higher probability) and the the top 3 extracted modes (blue arrows) for each method.

# B  Pseudocode

---

*Mixture Density Particle Filtering:*

Given observations $y_{1:T}$ and actions $a_{1:T}$ with $N$ being the number of particles

1. Initialize particle set $\{x_1^{(:)}, w_1^{(:)}, \beta\}$ using initial known state or via some other method. Normalize weights such that $\sum_{i=1}^{N} w_1^{(i)} = 1$

2. For $t = 2, ..., T$ **and** $i = 1, ..., N$

    (a) Resample particle from mixture distribution
    $$\widetilde{x}_t^{(i)} \sim m(x_{t-1}^{(:)}, w_{t-1}^{(:)}, \beta), \qquad \widetilde{w}_t^{(i)} = \frac{1}{N}$$

    (b) Apply noisy system dynamics to particle:
    $$x_t^{(i)} = f(\widetilde{x}_t^{(i)}, a_t, \eta), \qquad \eta \sim N(0, 1)$$

    (c) Compute particle weight (normalized such that $\sum_{i=1}^{N} w_t^{(i)} = 1$)
    $$w_t^{(i)} = \widetilde{w}_t^{(i)} \cdot l(x_t^{(i)}; y_t)$$

3. **Output:** $\{x_{1:T}^{(:)}, \widetilde{w}_{1:T}^{(:)}, w_{1:T}^{(:)}, \beta\}$

---

Figure 12: The Mixture Density Particle Filter

---

*Mixture Density Particle Smoothing:*

Given observations $y_{1:T}$ and actions $a_{1:T}$ with $N$ being the number of particles

1. Compute forward filter particle sets using Mixture Density Particle Filtering with $y_{1:T}$ and $a_{1:T}$:
$$\{\overrightarrow{x}_{1:T}^{(:)}, \overrightarrow{\widetilde{w}}_{1:T}^{(:)}, \overrightarrow{w}_{1:T}^{(:)}, \overrightarrow{\beta}\}$$

2. Compute backward filter particle sets using Mixture Density Particle Filtering with $y_{T:1}$ and $a_{T:1}$ (with time reversed):
$$\{\overleftarrow{x}_{1:T}^{(:)}, \overleftarrow{\widetilde{w}}_{1:T}^{(:)}, \overleftarrow{w}_{1:T}^{(:)}, \overleftarrow{\beta}\}$$

3. For $t = 1, ..., T$ **and** $i = 1, ..., N$

    (a) Define:
    $$q(x) = \frac{1}{2} m(x; \overrightarrow{x}_{1:T}^{(:)}, \overrightarrow{\widetilde{w}}_{1:T}^{(:)}, \overrightarrow{\beta}) + \frac{1}{2} m(x; \overleftarrow{x}_{1:T}^{(:)}, \overleftarrow{\widetilde{w}}_{1:T}^{(:)}, \overleftarrow{\beta})$$

    (b) Sample particles:
    $$\overleftrightarrow{x}_t^{(i)} \sim q(x)$$

    (c) Compute weights
    $$\overleftrightarrow{w}_t^{(i)} = \frac{l\left(\overleftrightarrow{x}_t^{(i)}; y_t, m(\overleftrightarrow{x}_t^{(i)}; \overrightarrow{x}_{1:T}^{(:)}, \overrightarrow{\widetilde{w}}_{1:T}^{(:)}, \overrightarrow{\beta}), m(\overleftrightarrow{x}_t^{(i)}; \overleftarrow{x}_{1:T}^{(:)}, \overleftarrow{\widetilde{w}}_{1:T}^{(:)}, \overleftarrow{\beta})\right)}{q(\overleftrightarrow{x}_t^{(i)})}$$

4. **Output:** $\{\overleftrightarrow{x}_{1:T}^{(:)}, \overleftrightarrow{w}_{1:T}^{(:)}, \overleftrightarrow{\beta}\}$

---

Figure 13: The Mixture Density Particle Smoother

# C Number of Particles Ablation Study

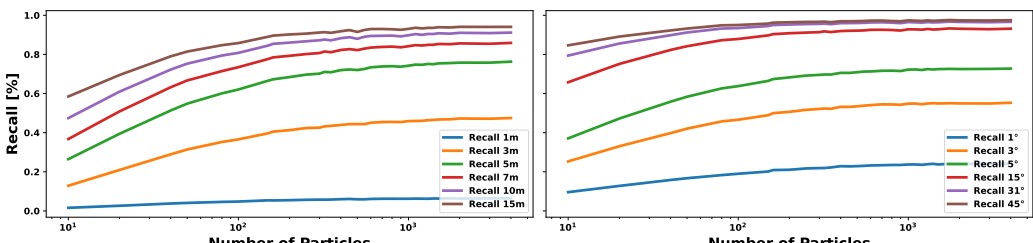

Figure 14: Recall of position and angle for MDPS using the MGL dataset [11] with varying numbers of particles at inference time. Here we specify the number of particles $N$ used for the forward and backward filters of MDPS. The final MDPS posterior density is defined by $2N$ particles as described in sec. 4. Interestingly, performance plateaus quickly as we increase the number of particles implying MDPS's ability to use particles smartly and efficiently, allowing for fewer particles to be used, lowering the computation and memory requirements neeeded for effective models.

A key hyper-parameter for particle filters and smoothers is the number of particles to use at inference time. Using more particles increases performance as shown in fig. 14 but performance can quickly plateau. As seen in fig. 4, effective models tend to concentrate particles densely in likely regions of the state space. By using more particles, the particle density of these likely regions is increased but with diminishing returns. Each additional particle within the dense regions will vary only slightly from its neighbors, minimally adding to the particle set diversity. Further using more particles increases computation and memory requirements making smarter models which are more particle efficient, such as our MDPS, attractive for real world deployment

## D   Additional Experiment Details

### D.1   Bearings Only Tracking Task

The Bearings Only Tracking Task adopted from Younis and Sudderth [14] aims to track the state of a vehicle as it navigates a simple environment. No actions are provided for this task and the observations are given as noisy bearings to a radar station:

$$y_t \sim \alpha \cdot \text{Uniform}(-\pi, \pi) + (1 - \alpha) \cdot \text{VonMises}(\psi(x_t), \kappa),$$

where $\psi(x_t)$ is the true bearing with $\alpha = 0.15$ and $\kappa = 50$. The velocity of the vehicle varies over time, changing randomly when selecting a target new way point with 1m/s or 2m/s being equally likely. During training, ground truth states are provided every 4 time-steps however dense true states are given at evaluation time. All sequences are of length $T = 50$ and we use 5000, 1000 and 5000 sequences for training, validation and evaluation respectively.

For all methods we use 50 particle during training and evaluation. For forward-in-time running PF methods, we initialize the particle set as the true state with small Gaussian noise ($\sigma = 0.01$) on the x-y components of the state. For the angle components of the initial particles we add Von Mises noise (with concentration $\kappa = 100$) to the true state. For backward-in-time running PF methods we set the initial particle set as random samples drawn uniformly from the state space.

Learning rates are varied throughout the training stages ranging from 0.001 to 0.000001 though we find that all methods are robust to learning rate selection when using the Adam [49] optimizer, with sensible learning rate effecting convergence speed but not performance.

For SR-PF [13] we set $\lambda = 0.1$.

#### D.1.1   Model Architectures

All methods (baselines and ours) for the Bearings Only Tracking Task use the same dynamics and measurement model architectures which are described below.

**Dynamics Models.** We parameterize the dynamics model as a residual neural network as shown in figs. 15 and 16. To maintain position in-variance, we mask out the position components of particles when input into the dynamics model. We also transform the angle component of the state into a vector representation $T(\theta) = (\sin(\theta), \cos(\theta))$ before applying dynamics. Afterwards we transform the angle component of the state back into an angle representation $T(u, v) = \text{atan2}(u, v) = \theta$

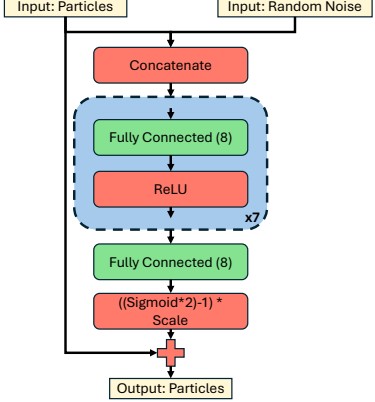

Figure 15: Dynamics model used for the Bearings Only Tracking Task. The output scaling scales the position components of the residual to be within $[-5, 5]$ and $-2, 2$ for the positional and angle (in vector representation) components respectively.

**Measurement Models.** Figure 17 shows the feed-forward neural network architecture for the measurement models used for the Bearings Only Tracking Task.

**MDPS Forward Backward Combination.** For the Bearings Only Tracking Task, the MDPS smoothed measurement model is very similar to the measurement model using for MDPF but with additional inputs. The MDPS smoothed measurement model network architecture is shown in fig. 18.

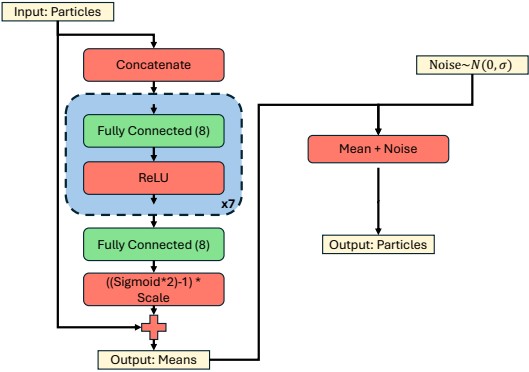

Figure 16: Dynamics model used for FFBS in the Bearings Only Tracking Task. The output scaling scales the position components of the residual to be within $[-5, 5]$ and $-2, 2$ for the positional and angle (in vector representation) components respectively. The dynamics model outputs a mean. Using this mean along with a hand tuned standard deviation allows for simulation from the dynamics model as well as evaluating state transition probabilities as required for FFBS. Of note: the angle dimension of the state is approximated by a Normal distribution with bound variance to avoid issues with angular discontinuities. The hand-tuned standard deviations values used are $[1.0, 1.0, 1.25]$.

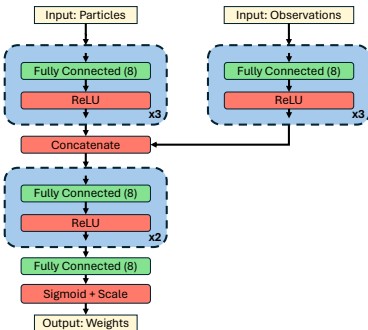

Figure 17: Particle filter measurement model used for the Bearings Only Tracking Task. The output scaling scales the weights to be within $[0.00001, 1]$

## D.2 Global Localization Task with Mapillary Geo-Location Dataset and KITTI Datasets

In this section we give more information about the experiments conducted with the Mapillary Geo-Location Dataset (MGL)[11] and KITTI [51] datasets.

For all particle filter methods (including ones internal to MDPS) we use 250 particle during training and evaluation and initialize the filters using 1000 particles. For PF and smoother methods, we initialize the particle set as the true state with Gaussian noise ($\sigma = 50$ meters) on the x-y components of the state. For the angle components of the initial particles we add Von Mises noise to the true state.

Initial learning rates are varied throughout the training stages ranging from 0.01 to 0.000001 though we find that all methods are robust to learning rate selection when using the Adam [49] optimizer, with sensible learning rates effecting convergence speed. For comparison methods, we use the learning rates as specified by the method authors or select them via a brief hyper-parameter search if they are not stated.

### D.2.1 Mapillary Geo-Location Dataset Additional Details

In the MGL dataset, observations are images captured by various types of handheld or vehicle mounted cameras. Some cameras capture $360°$ images which require additional processing before being used as observations. These $360°$ images are cropped to a $90°$ Field-of-View in random viewing directions, with the same viewing direction being used for the whole observation sequence. All images are then gravity aligned to produce the observation sequence. A planimetric map of the

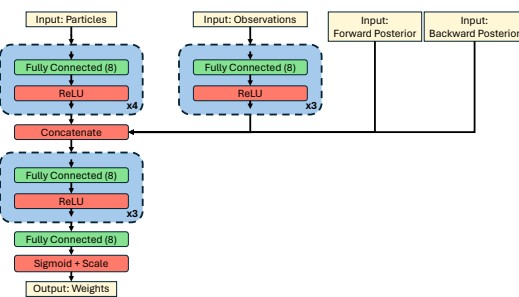

Figure 18: Measurement model used for the Bearings Only Tracking Task when computing the smoothed particle weights for MDPS. The output scaling scales the weights to be within $[0.00001, 1]$

environment is also provided via the OpenStreetMap platform [54] at 0.5 meter/pixel resolution, and all observation images are publicly available under a CC-BY-SA license via the Mapillary platform. All KITTI data is published under the CC-BY-NC-SA licence.

Unfortunately the creators of the MGL dataset trained their methods on single observations from the dataset and did not use sequences during training [11]. As such, observations from sequences are scattered amongst the training and testing splits, preventing effecting training of methods that require longer uninterrupted sequence data such as MDPF and MDPS. We therefore create custom train, validation and evaluation splits of the MGL dataset in order to accommodate longer sequences for training and evaluation.

Due to data integrity and corruption issues we exclude all sequences from the "Vilnius" portion of the dataset.

### D.2.2 MDPF/MDPS Model Architectures

**Dynamics Models.** The network architecture of the dynamics model used for the MGL and KITTI is shown in fig. 19 and is similar to that used in the Bearings Only Tracking task, using the same angle to vector particle transformation. Note for this dynamics model we do not mask out any component of the state.

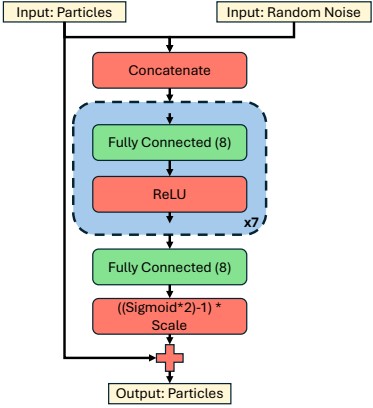

Figure 19: Network architecture for the dynamics model used for the Bearings Only Tracking Task. The output scaling scales the position components of the residual to be within $[-128, 128]$ and $-2, 2$ for the positional and angle (in vector representation) components respectively.

**Measurement Models.** The measurement model used for MDPF and MDPS uses the Birds-Eye-View (BEV) feature encoder and the map encoder from the official Dense Search implementation released by Sarlin et al. [11]. As shown in fig. 1, a Birds-Eye-View (BEV) feature map is estimated from the observation using a geometric projection [11] where columns of the observation image are considered polar rays with features binned into course depth planes projected away from the camera focal plane. This gives a top-down representation of the local area in polar coordinates (bearing and

course distance of an image feature from the camera center). The polar representation of the scene is then sampled into top-down Cartesian coordinates to yield the final BEV feature map. We refer the reader to Sarlin et al. [11] and the appendix for more details. To compute particle weights, we compute the alignment, via a dot-product, between the BEV feature map and a local region from the neural map, cropped and rotated at the current particles location. For $l(\cdot)$ from eqn. 20 we use fully-connected layers to produce the final smoothed particle weights from the BEV-map dot-product alignment and the forward and backward filter posterior densities. The map feature encoder is a U-Net based architecture with a VGG-19 [56] backbone and the BEV feature encoder is based on a multi-head U-Net with a ResNet-101 [57] backbone as well as a differentiable but un-learned geometric projection. We refer the reader to Sarlin et al. [11] for more details about the encoders. To derive the un-normalized particle log-weights, the BEV encoding is compared, via dot-product, to a local map patch extracted at a specific particle to compute an alignment value. This is akin to the dense search described in Sarlin et al. [11] but at only a single location determined by the particle.

**MDPS Forward Backward Combination.** The MDPS smoother measurement model differs from the measurement models of MDPF as it requires additional inputs as described in eqn 20. We implement this model as a 4-layered, 64-wide fully-connected feed-forward neural network with PReLU [58] activation's shown in fig. 20. Importantly all computed un-normalized weights are bound to be within $[0.0001, 1]$ using a Sigmoid function with an offset. Input into this network is the BEV-map feature alignment computed in the same way as the MDPF measurement model as well as the posterior probability values from the forward and backward filters for the current smoothed particle.

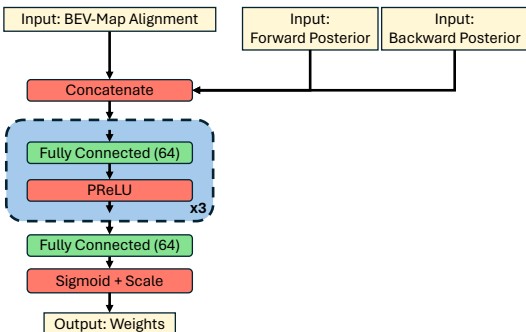

Figure 20: Network architecture for the MDPS smoothed measurement model used for the MGL and KITTI datasets.

### D.2.3   MDPS Training Procedure Details

As stated in sec. 4, effective training of MDPS requires training in stages. Importantly due to VRAM constraints, we are unable to train large map and observation encoders on long sequences. Therefore we train on short sequences before freezing the encoders, training the rest of the models on longer sequences.

Training procedure for MDPS on the MGL dataset:

1. Train forward and backward MDPFs individually via independent loss functions, sharing map and observation encoders, on short sequences from the dataset. Here we hold the output posterior bandwidths fixed to prevent converging to poor local optima where the bandwidth is widened while the dynamics and measurement models are not informative.

2. Freeze all MDPF models, unfreeze the MDPF output posterior bandwidths and train on long sequences.

3. Freeze all MDPF models (including bandwidths) and train the MDPS measurement model and output posterior bandwidth on long sequences.

4. Unfreeze all models except the map and BEV encoders and train MDPS on long sequences.

Due to the small size of the KITTI dataset along with pre-training using the MGL dataset, a special constrained training procedure is required to prevent immediate over-fitting to the training split.

Instead of jointly refining all components simultaneously, we fine-tuning the forward and backward MDPFs before freezing those models and fine-tuning the MDPS smoothed measurement model:

1. Train the forward and backward MDPFs individually via independent loss functions, sharing map and observation encoders, on short sequences from the dataset. We hold the output posterior bandwidths fixed.

2. Freeze BEV and map encoders, training the forward and backward MDPFs individually via independent loss functions on long sequences.

3. Freeze all MDPF models, unfreeze the MDPF output posterior bandwidths and train on long sequences.

4. Freeze all MDPF models (including bandwidths) and train the MDPS smoothed measurement model and output posterior bandwidth on long sequences.

5. Freeze all models except for the MDPS output posterior bandwidth and train on long sequences.

### D.2.4 Baseline Implementation Details

**Retrieval.** The Retrieval [3] baseline as described by Samano et al. [3] encodes individual patches from the environment global map into the a latent space. This is inefficient if the map patches are densely sampled and is prohibitive to run for large environments. Instead a dense feature map can be predicted from the global map in one forward CNN pass to generate dense map encoding for map patches roughly sized according to the CNN receptive field [11]. We adopt this dense encoding approach for Retrieval, implementing the Retrieval method as specified by Sarlin et al. [11].

**Dense Search.** For Dense Search we use the official implementation released by Sarlin et al. [11] which differs from the text description present in the official paper. In the paper description a location prior is computed from the provided map which estimates regions of the state space that are likely to be occupied (e.g. the prior says that rivers and inside buildings are unlikely to be occupied by a car). The prior is then multiplied with the observation likelihood (probability volume computed via dense search) to produce the final state posterior. In the official implementation this prior is disabled making the state posterior simply the observation likelihood. Further we use VGG-19 [56] as our map encoder backbone.

**Retrieval (PF).** Zhou et al. [8] embeds standard Retrieval methods within a non-differentiable particle filter where the dynamics are set as Gaussian Noise:

$$x_t^{(i)} \sim \mathcal{N}(x_{t-1}^{(i)} + a_t, \gamma) \tag{24}$$

In our experiments we set $\gamma$ as $2.5m$ for the x-y state position components and $15°$ for the angular state components, chosen via a brief hyper-parameter search. The measurement model is defined as

$$w_t^{(i)} = \exp\left(\frac{-d_t^{(i)}}{2\sigma^2}\right) \tag{25}$$

where $d_t^{(i)}$ is the alignment of the observation latent encoding with the map patch encoding at the current particles location $x_t^{(i)}$,computed via the Retrieval method. After a brief hyper-parameter search we set $\sigma = 2$ in our experiments.

**Applying baselines to city-scale environments.** Due to memory constraints, dense search over the whole map is not possible (approx. 809 GB is needed for $T = 100$ length sequence at 0.5m per pixel resolution). We therefore offer 2 methods for applying this dense search at city scales. *Ground Truth (GT) Cheat Method:* using the ground truth state, we extract a small region from the map in which we do dense search. This greatly reduces the search space, saving memory but also greatly (and *artificially*) improves performance. *Sliding Window Method:* At $t = 1$ a small region extracted around the true state is densely searched. At subsequent time-steps, the best alignment from the dense search of the previous time-step is propagated using $a_t$ and used as the center of a new small region which is then searched. Retrieval methods tend to fail when applied to large environments with Zhou et al. [8] even limiting the search to patches on known road networks. To address this, we limit the search space of Retrieval like in Dense Search using the GT Cheat and Sliding Window techniques, considering map patches within small regions.

### D.2.5 Top 3 Mode Finding via Non-Maximal Suppression

Due to multi-modality of the posterior estimate, simply extracting the top mode to evaluate errors is not a good gauge of performance. Instead we to extract the top-3 modes from the posterior density for evaluation. This can be easily achieved via a non-maximal suppression scheme where modes are extracted before particles around those modes are deleted. Specifically after extracting the top mode, the distance of all particle to that top mode is calculated. Particle within some threshold of the top mode are deleted from the particle set and the weights of all remaining particles are re-normalized to admit a valid probability density after deletion. The next top mode is then extracted and the deletion process repeated until a total of 3 modes are extracted. In our implementation, we delete particles that are within 5m and $30°$ of the top mode.

For methods that admit a discrete probability volume (such as Dense Search and Retrieval), we suppress the values of all probability cells within 5m and $30°$ of the top mode during the deletion step. Extracting the top mode can be simply achieved by finding the maximum value within the discrete probability volume.

### D.3 Compute Resources

We give an approximation for compute resources needed to run our experiments in tables 1, 2 and 3. Since our experiments are bottle-necked by GPU resources and require only minimal CPU and memory needs, we report the GPU needs and GPU runtime for each experiment. In addition to the compute resources stated in this section, we used additional resources over the course of our project when developing our methods, though the amount of resources used is difficult to quantify and thus we do not report here.

For evaluation, all methods can comfortably run in under 2 hours for the full evaluation split of MGL (using a NVIDIA A6000 GPU) and in under 1 hour for Bearings Only (using a NVIDIA RTX 3090 GPU) and KITTI (using a NVIDIA A6000 GPU).

Table 1: Computation needs for training Bearing Only Tracking Task. Of note: none of the methods require using the whole GPU and thus we usually train 2-3 methods per GPU simultaneously. The numbers reported assume training each method 11 times sequentially without running in parallel.

| Experiment | GPU | GPU Runtime |
|---|---|---|
| TG-PF (Multinomial) | 1x NVIDIA RTX 3090 | $\sim$25 hrs |
| TG-PF (Stratified) | 1x NVIDIA RTX 3090 | $\sim$25 hrs |
| SR-PF (Multinomial) | 1x NVIDIA RTX 3090 | $\sim$21 hrs |
| SR-PF (Stratified) | 1x NVIDIA RTX 3090 | $\sim$21 hrs |
| MDPF (Multinomial) | 1x NVIDIA RTX 3090 | $\sim$80 hrs |
| MDPF (Stratified) | 1x NVIDIA RTX 3090 | $\sim$80 hrs |
| MDPF-Backward | 1x NVIDIA RTX 3090 | $\sim$80 hrs |
| MDPS | 1x NVIDIA RTX 3090 | $\sim$160 hrs |

Table 2: Computation needs for global localization on the MGL dataset. Retrieval (PF) requires no training since all trained models are taken from the Retrieval baseline. Similarly MDPF requires no training as it is trained within MDPS.

| Experiment | GPU | GPU Runtime |
|---|---|---|
| Retrieval | 1x NVIDIA A6000 | $\sim$12 hrs |
| Retrieval (PF) | – | NA (No Training) |
| Dense Search | 4x NVIDIA A6000 | $\sim$48 hrs |
| MDPF | – | NA (No Training) |
| MDPS | 3x NVIDIA A6000 | $\sim$72 hrs |

Table 3: Computation needs for global localization on the KITTI dataset. Retrieval (PF) requires no training since all trained models are taken from the Retrieval baseline. For MDPF we report additional resources used during refinement on the KITTI dataset.

| Experiment | GPU | GPU Runtime |
|---|---|---|
| Retrieval | 1x NVIDIA A6000 | ~1 hrs |
| Retrieval (PF) | – | NA (No Training) |
| Dense Search | 4x NVIDIA A6000 | ~18 hrs |
| MDPF | 3x NVIDIA A6000 | ~5 hrs |
| MDPS | 3x NVIDIA A6000 | ~10 hrs |

