# OpenReview forum: "Learning to be Smooth: An End-to-End Differentiable Particle Smoother"
_NeurIPS.cc/2024/Conference — NeurIPS 2024 poster_

### Official Review · Reviewer_56kY · 2024-06-22

**Soundness:** 3
**Presentation:** 2
**Contribution:** 3
**Rating:** 7
**Confidence:** 4

**Summary:**

This paper proposes an end-to-end differentiable particle smoother and presents its application to global localization of a subject moving through real-world city-scale environments. Although there exists prior work in differentiable particle *filtering*, such as Mixture Density Particle Filters (MDPF), *smoothing* yields a more accurate estimate for offline data processing as it incorporates all the observations taken along the full trajectory. This paper revisits the classic formulation of Two Filter Smoothing (TFS) and leverages its principle to fuse two MDPFs, one running forward in time and one backward, into the novel Mixture Density Particle Smoother (MDPS). MDPS achieves higher log-likelihood and better tracking accuracy compared to MDPF and other prior work.

**Strengths:**

* To my knowledge, this paper proposes the first framework for end-to-end differentiable particle smoothing. Smoothing is a powerful tool for offline state estimation and expected to achieve better accuracy than filtering. The proposed MDPS is a valuable framework for offline state estimation that does not require hand-designed state transition or observation models.

* The technical approach is a sound combination of MDPF and TFS. Although I have some remaining questions about the underlying math (which is discussed below), the authors seem to have successfully designed their differentiable smoothing framework by leveraging the principled TFS formulation.

* Empirical results show a clear advantage of the smoothing framework compared to its filtering counterpart as well as other prior work based on non-probabilistic schemes (e.g. refinement).

**Weaknesses:**

[EDIT] All major concerns have been resolved after the rebuttal.

* Some of the math presented in Section 3 for TFS seems to be incorrect. Specifically, I do not think equation (14) holds; it seems to be missing $p(y_t \mid x_t)$ from its numerator. Similarly, the resulting equation (15) for the particle weights does make sense either; multiplying a weight by a weighted sum of estimated states would not result in a weight. It would instead result in a new state. Please check (14) and (15) again and correct any errors. Also, please clarify how (14) and (15) relate to the practical implementation of MDPS presented in Section 4. In its current form of the paper, (14) and (15) seem unnecessary for MDPS because we instead have the correct equation (16) to begin with.

* Many figures should be updated for better readability when printed on paper. Figures 1, 4, 7 - 11 have map patches and RGB observations that are too small. I suggest that the authors increase the font size as well and make each component in the figures bigger. The same goes for Figures 3 and 6, where the legends are too small to associate different curves with different methods.
More details on MDPF would be appreciated at the end of Section 2 for self-completeness of the information. Specifically, I did not fully follow what the “adaptive” variant means and why two bandwidths was necessary.

* More details on MDPF would be appreciated at the end of Section 2 for clarity. Specifically, I did not fully follow what the “adaptive” variant means and why two bandwidths were necessary.

**Questions:**

[EDIT] All major concerns have been resolved after the rebuttal.

* For MDPS, did you ever try initializing the estimates from some random states rather than the noised version of the true states? I wonder whether smoothing can actually correct erroneous initial estimates. If true, that is another advantage compared to MDPF, since we cannot hope for filtering to fix errors in initial estimates at all.

* MDPS requires more GPU runtime to train than MDPF by design, as presented in Tables 1, 2, and 3. Would MDPF still perform worse than MDPS even when you let MDPF train longer to match the GPU runtime of MDPS? In other words, would the loss and the accuracy of MDPF not improve further with additional training epochs?

**Limitations:**

* As a fundamental limitation of the proposed approach, training requires three separate stages and thus takes more GPU runtime. I would appreciate it if the authors could comment on the tradeoff between better performance and more demand for compute when compared to MDPF.

---

> ### Author Rebuttal · Authors · 2024-08-06
>
> Thank you for the positive comments and constructive feedback.
>
> Please see our global response to all reviewers for discussion of training time and paper figure formatting.
>
> We agree that adding some discussion of accuracy vs compute demands of MDPF and MDPS would be useful, and will do this. In general MDPS will have higher compute demands than MDPF due to running two particle filters, but both methods scale similarly with the number of particles, and our experiments show that the constant-factor cost increase of MDPS may substantially boost accuracy. Note that appendix B discusses accuracy vs. compute demands for MDPS with respect to the number of particles used.
>
> We would like to thank you for taking the time to carefully review our math. Indeed there are factors missing from equations (14) and (15), below are the corrected versions (with the corrections highlighted in red):
>
> **Equation (14):**
>
> $p(x_t|y_{1:T}) \propto \frac{\tilde{p}(x_t| y_{t+1:T}) \color{red} p(y_t|x_t) \color{black} \int p(x_t|x_{t-1}) p(x_{t-1} | y_{1:t-1}) dx_{t-1}} {\gamma_t(x_t)}$
>
> **Equation (15):**
>
> $\overleftrightarrow{w}_{t}^{(i)} \propto \overleftarrow{w}\_{t}^{(i)} \sum\_{j=1}^N  \color{red} \overrightarrow{w}\_{t-1}^{(j)} \color{black}  \frac{p(\overleftarrow{x}\_{t}^{(i)} | \overrightarrow{x}\_{t-1}^{(j)})}{\gamma_t(\overleftarrow{x}\_{t}^{(i)})}$
>
> Equations (14) and (15) are included in the paper to contrast our MDPS formulation with the two-filter formulation employed by most prior work. You are correct that they are not directly used in the derivation of MDPS in Sec. 4, and thus these missing terms do not impact the correctness of our results.
>
> We agree that the details on MDPF in Sec. 2 are brief, and that expanding this section could be beneficial.  In short, the adaptive MDPF as described in [14] is simply a particle filter that decouples the distribution used for resampling and the distribution used as the posterior estimate. The rationale is that the distribution used for resampling particles may not necessarily be appropriate for estimating the state posterior, since the latter may concentrate probability mass more tightly when observations are informative. Adaptive MDPF decouples the resampling and posterior bandwidths and learns each separately, allowing the resampling and posterior distributions to be separately tuned for each application.
>
> **To answer your specific questions:**
>
> - With regard to MDPS initialization, in the Mapillary and Kitti experiments we initialize MDPS with particles within approximately 150 meters of the true state with the localization area only being 614x614 meters (i.e. particles are drawn randomly from about ~19% of the localization area). This initialization results in erroneous initial estimates for MDPF but interestingly we find that MDPS is able to correct these erroneous initial estimates by using information from the backward filter. Given this, we believe that MDPS could correct initial states when initialized with totally random particles.
>
> - Please see our global response to all reviewers for discussion of training time.

---

> > ### Comment · Reviewer_56kY · 2024-08-10
> > **Response to Authors**
> >
> > I greatly appreciate the authors for taking the time and effort to prepare the rebuttal. The initial response by the authors have largely addressed my concerns, and I will increase my score accordingly. Below please find my response.
> >
> > * > We agree that the details on MDPF in Sec. 2 are brief, and that expanding this section could be beneficial. In short, the adaptive MDPF as described in [14] is simply a particle filter that decouples the distribution used for resampling and the distribution used as the posterior estimate. The rationale is that the distribution used for resampling particles may not necessarily be appropriate for estimating the state posterior, since the latter may concentrate probability mass more tightly when observations are informative. Adaptive MDPF decouples the resampling and posterior bandwidths and learns each separately, allowing the resampling and posterior distributions to be separately tuned for each application.
> >
> >   Thank you for clarifying. Please include this piece of information into the paper if the space permits.
> >
> > * > With regard to MDPS initialization, in the Mapillary and Kitti experiments we initialize MDPS with particles within approximately 150 meters of the true state with the localization area only being 614x614 meters (i.e. particles are drawn randomly from about ~19% of the localization area). This initialization results in erroneous initial estimates for MDPF but interestingly we find that MDPS is able to correct these erroneous initial estimates by using information from the backward filter. Given this, we believe that MDPS could correct initial states when initialized with totally random particles.
> >
> >   This seems to be a reasonable performance improvement to expect, over any filtering-based approaches. I highly recommend the authors to highlight this point in the paper so the fundamental advantage of smoothing is made even clearer.

---

### Official Review · Reviewer_fbEx · 2024-07-10

**Soundness:** 1
**Presentation:** 1
**Contribution:** 1
**Rating:** 3
**Confidence:** 3

**Summary:**

My understanding is that this work seeks to learn a state-space model by differentiating through a regularised version of the sequential Monte Carlo (SMC) approximation of the the (generalised) two-filter smoother.

**Strengths:**

I am not very familiar with the applications used as numerical examples but they seem to be reasonably close to real-world scenarios.

**Weaknesses:**

**Lack of clarity & rigour**

I do not believe that the presentation of this work is adequate for NeurIPS.  The manuscript is much too informal and therefore very difficult to follow and verify. It reads like a mid-1990s engineering conference paper. For example,

* The work should really include some pseudo-code. Without this, it is very hard to even understand what the proposed algorithm exactly is. It is not even entirely clear to me whether the proposed method seeks to learn only the measurement model or also the system equation. The short paragraph on "Training details" at the end of Section 4 is also highly insufficient.
* Throughout Section 3 and 4, the same symbols seem to be used for model densities and their (e.g. Monte Carlo or other) approximations
* Lines 70/71 claim: "Our differentiable PS incorporate stratified resampling to reduce variance with negligible computational overhead, making training more robustly avoid local optima". I don't find evidence for this and it is not clear to me how the authors would achieve differentiable stratified resampling.
* Some notation is not or insufficiently clearly defined, e.g. $\hat{w}^{(i)}$ and its gradient in Equation 7.
* Line 108 mentions "unbiased" gradient estimates but I cannot see how this would be true for any of the proposed methodology.

**Missing baselines**

There are well-known SMC based, recursive parameter-estimation methods against which the method should really be compared, e.g.:

Kantas, N., Doucet, A., Singh, S. S., Maciejowski, J., & Chopin, N. (2015). On particle methods for parameter estimation in state-space models.

**Insufficient motivation**

It is not clear to me what the motivation for the proposed approach is, except that no one has previously tried to differentiate through the two-filter smoothing approximation.

**Questions:**

Can you explain how the proposed method achieves differentiable stratified resampling?

**Limitations:**

I appreciate the authors' candour that the proposed method will suffer from a curse of dimension.

---

> ### Author Rebuttal · Authors · 2024-08-06
>
> Thank you for your comments and feedback.
>
> **To address your concerns:**
>
> - Our MDPS method learns both the measurement model and the dynamics model via end-to-end training.  The “Training Details” paragraph at the end of section 4 was intended to highlight the 3-stage training approach. Additional training details for reproducibility can be found in the appendix.
>
> - We apologize that aspects of the notation were confusing, we will add more clarifications.  With regard to $\hat{w}^{(i)}$ in Eq. (7), this refers to the unnormalized weight assigned to the resampled particle $\hat{x}^{(i)}$.  It has weight 1 for the current model parameters, but weights must be adjusted as dynamics and observation parameters change during training, as captured by the gradient equation.
>
> - Line 108 mentions how unbiased gradient estimates may be produced via the Importance Weighted Sample Gradients (IWSG) estimator proposed in [14].  IWSG computes gradients based on importance sampling [21]. Briefly, importance sampling itself produces unbiased weighted samples from some distribution of interest, and therefore differentiating through importance sampling yields unbiased gradients as well.  Experiments in [14] show that this approach has dramatically reduced variance compared to reparameterization-based gradient estimators.
>
>
> - Thank you for pointing out the review of classic work by Kantas et al. (2015), we will add citation and discussion.  These parameter estimation techniques make different assumptions than our work:  they learn generative rather than discriminative particle filters, and assume that parametric dynamics and likelihood models with only a few unknown parameters are available.  (Experiments in Kantas have only 3 unknown parameters, and we are not aware of any classic work with more than a few dozen learned parameters.)  These methods would not practically scale to datasets with large numbers of sequences as we consider, because particle state estimates must be produced for every time step at every iteration.  Moreover, the neural network parameterizations of our dynamics and measurement models have over 500 million parameters, millions of times larger than work cited by Kantas.  The computational cost of classic particle learning algorithms scales poorly with the number of parameters, and it would be practically impossible to apply these methods to our models (without substantial innovations).  Indeed, these limitations motivated our research on the MDPS method.
>
>
> **To answer your specific questions:**
>
> - Stratified resampling [31,32] is a method of drawing i.i.d. samples from some target distribution while reducing the variance associated with those samples.  These samples can be thought of as uniformly weighted samples from the target distribution (i.e. where all the weights are equal). To compute gradients for these samples we use Importance Weighted Sample Gradients (IWSG) [1] which can be applied to any sampling method that produces i.i.d. samples. IWSG propagates gradients via adjusting the weights of the drawn samples (which are initially uniform), yielding a differentiable stratified resampling technique that has the lower sampling variance properties of stratified resampling and the unbiased gradients of IWSG.
>
> **Additional Reference:**
>
> Kantas, N., Doucet, A., Singh, S. S., Maciejowski, J., & Chopin, N. (2015). On particle methods for parameter estimation in state-space models.

---

> ### Comment · Reviewer_fbEx · 2024-08-12
> **Proof of unbiasedness**
>
> > Line 108 mentions how unbiased gradient estimates may be produced via the Importance Weighted Sample Gradients (IWSG) estimator proposed in [14]. IWSG computes gradients based on importance sampling [21]. Briefly, importance sampling itself produces unbiased weighted samples from some distribution of interest, and therefore differentiating through importance sampling yields unbiased gradients as well. Experiments in [14] show that this approach has dramatically reduced variance compared to reparameterization-based gradient estimators.
>
> Thank you. I have now read [14]. It seems to me that the IWSG from [14] approximates the particle approximation of each filter by a kernel density estimate which is indeed differentiable.
>
> However, due to the introduction of this extra layer of (kernel-density) approximation,  I don't see how the IWSG method from [14] would give unbiased gradients of the **original** particle filter. It might give unbiased gradients of some (likely biased) approximation. But **having unbiased gradients for an approximation of a particle filter is not the same as having unbiased gradients for the particle filter itself.**
>
> Unfortunately, like the present manuscript, [14] is written in a highly informal manner and does not seem to provide any proof of the claimed unbiasedness. I would be willing to raise my score if the authors could actually provide a proof of the claim that the proposed method from [14] gives unbiased gradients for the **original** particle filter. In my view, providing rigorous proofs of such claims is a must for a NeurIPS publication.

---

> > ### Comment · Reviewer_fbEx · 2024-08-12
> > **Unbiased gradients of stratified resampling**
> >
> > > Stratified resampling [31,32] is a method of drawing i.i.d. samples from some target distribution while reducing the variance associated with those samples. These samples can be thought of as uniformly weighted samples from the target distribution (i.e. where all the weights are equal). To compute gradients for these samples we use Importance Weighted Sample Gradients (IWSG) [14] which can be applied to any sampling method that produces i.i.d. samples. IWSG propagates gradients via adjusting the weights of the drawn samples (which are initially uniform), yielding a differentiable stratified resampling technique that has the lower sampling variance properties of stratified resampling and the unbiased gradients of IWSG.
> >
> > To raise my score, I would want to see an actual proof that the claim "IWSG gives unbiased estimates of (stratified) resampling" is true. As discussed in my previous comment, what the present method at most gives is an unbiased estimate of a gradient of some (very likely biased) **approximation** of the particle filter (for any arbitrary resampling scheme).

---

> > > ### Author Response · Authors · 2024-08-12
> > > **Re: Unbiased gradients of stratified resampling**
> > >
> > > We emphasize that the “true” particle filter that we are trying to approximate, which will be run at test-time, is a “regularized” particle filter that uses KDEs to approximate marginals.  Please see our other comment (Re: Proof of unbiasedness) for more details on this point.
> > >
> > > For the KDE resampling step, some number of particles will be drawn from each of the KDE mixture components, before continuous sampling from that component’s kernel (a Gaussian or von Mises density in our experiments).  While stratified resampling allocates particles to mixture components via a distinct algorithm, the expected number of particles allocated per component is identical to classic multinomial resampling.  See [31, Sec. 2.3] for a proof of this.  Then by the linearity of the expectation operator, gradients of this stratified resampling step may be computed via the same importance-sampling estimator we use for multinomial resampling.
> > >
> > > We appreciate the question, and agree this point did not have adequate detail in our submitted manuscript.  We will add additional detail (adapting the proof in [31] to our setup/notation) in future revisions.

---

> > > > ### Comment · Reviewer_fbEx · 2024-08-14
> > > >
> > > > Thank you. I still don't exactly understand what the authors mean by "stratified" resampling in their context. But it is not what is normally called "stratified resampling" in the literature.
> > > >
> > > > I continue to think that this paper needs to be greatly improved in terms of rigour, clarity and presentation before it is publishable. I would recommend that the authors take a look at the presentation of particle filters in publications in high-ranking computational statistics journals over the past decade for examples on how to achieve this.

---

> > ### Author Response · Authors · 2024-08-12
> > **Re: Proof of unbiasedness**
> >
> > We did not emphasize this due to space constraints, but the theoretical foundations of KDE-based particle filters (which are standardly called “regularized” particle filters in the literature) are quite strong.  The cited paper by Musso et al. [40] gives an overview, and detailed results are developed by Le Gland and Oudjane (Annals of Applied Prob. 2004).  For conventional discrete-resampling particle filters, theoretical results show weak convergence to the optimal filter posterior as the number of particles becomes large.  Regularized particle filters have similar guarantees, but by directly approximating the posterior density, one can also obtain strong convergence (in variational distance or other Lp norms) to the optimal posterior.  Moreover, while standard particle filters require strong “mixing” assumptions on the dynamics to avoid degeneracy, regularized filters behave better (in both theory and practice) for low-noise dynamics due to the diversity provided by their continuous resampling step.
> >
> > By “original” particle filter, we assume you mean the basic “bootstrap” particle-filter that does discrete resampling with replacement.  Computing gradients for such particle filters is not a goal of our paper, because they are inferior (in both theory and practice) to regularized (KDE-based) particle filters.  At test time, our mixture density particle filters/smoothers employ KDE regularization, and thus this regularization must also be accounted for when computing gradients during training.  As detailed in [14, Sec. 4], the IWSG method does give an unbiased estimate of gradients for the regularized, KDE resampling step.  The mathematical proof is simple because we build on classic importance sampling principles, but it is nevertheless novel (in this context) and empirically effective.
> >
> > In the classic particle filtering literature, regularized particle filters were already motivated by having greater test-time stability/robustness, as well as superior theoretical guarantees.  One interesting conclusion of our work is that in addition to having these advantages, the smoothness of regularized particle filters also has the advantage of making accurate gradient estimation easier.  Experimentally, differentiable particle filters based on discrete-resampling perform dramatically worse than our approach (see the TG-PF and SR-PF baselines in Figure 2 of this submission, and the wider experimental comparison in [14, Figure 5]).
> >
> > **Reference:**
> >
> > Le Gland, F., & Oudjane, N. (2004). Stability and uniform approximation of nonlinear filters using the Hilbert metric and application to particle filters. The Annals of Applied Probability, 14(1), 144-187.

---

### Official Review · Reviewer_qoBM · 2024-07-12

**Soundness:** 3
**Presentation:** 3
**Contribution:** 4
**Rating:** 7
**Confidence:** 4

**Summary:**

The author's propose the first differentiable particle smoother, building on the MDPF of Younis and Sudderth which replaces the multinomial resampling step with a sampling from a KDE mixture. Their smoother is a two filter smoother where both the forwards and backwards filters are MDPFs. Instead of using the forwards filter to re-weight the particles from the backwards filter, as is conventional, particles are drawn directly from the mixture of the forwards and backwards posteriors. The author's test their smoother, first a toy bearings only tracking example. Then the a more complicated visual localisation task as well and the KITTI dataset. The smoother out performs the state-of-the-art filtering and search-based methods tested.

**Strengths:**

1. A differentiable particle smoother is the obvious next step to differentiable filtering, especially given much previous work trains their filters in an offline, batched way on with access to ground truth latent state in training. I.e. on smoothing problems. This is the first paper to successfully construct a differentiable smoother capable of learning the complete model.
2. The experiment results are very strong and give good credibility to the algorithm. The used scenario is considerably more complex than those often used to test algorithms in a similar class.
3. Using the direct product of the mixtures simplifies the calculations greatly.
4. The paper is well organised and easy to follow.

**Weaknesses:**

1. This method and it's predecessor the MDPF appear to enjoy less theoretical support than other differentiable particle algorithms such as \'{S}cibior and Wood's stop gradient resampling or Thornton, Corenflos et al.'s OT resampler. For example neither the gradient estimates nor state estimates appear to be consistent, or at least there is no proof they are.
2. Unlike the MDPF, this smoother requires the pointwise evaluation of the posterior density. This differs from the usual two filter smoother, I am unconvinced that the KDE approximation is accurate enough for this purpose. Some more analysis could be used to justify this approach.
3. This method has potential impact outside of just visual localisation, instead of several visual localisation experiments an example of a different problem could have boosted the apparent impact.
4. The text on the figures is far too small.

**Questions:**

None.

**Limitations:**

It is mentioned in the limitations that the curse of dimensionality may be lifted somewhat with smarter proposal. There is no evidence that this algorithm nor the MDPF is adept at learning a non-bootstrap proposal.

---

> ### Author Rebuttal · Authors · 2024-08-06
>
> Thank you for your praise and feedback. We are grateful for the careful review of our work and appreciate your highlights on the novel nature of our differentiable particle smoother, and the strength of our experimental results compared to scenarios considered by related work.
>
> **To address some of your concerns please see below:**
>
> - We agree that KDE density approximations may not always be accurate, but established theory [41] tells us that KDEs consistently approximate any smooth density with a sufficient number of samples, and our experiments suggest they can provide accurate approximations with practical numbers of particles.
>
> - We too agree that our method has impacts outside of just visual localization and plan to apply our techniques to other tasks in future work.  But we would like to emphasize that given the complexity of the real-world datasets considered in our experiments, achieving these advances to the state-of-the-art in visual localization already required substantial work.
>
> - Please see our global response for more information on figures and formatting.

---

> > ### Comment · Reviewer_qoBM · 2024-08-11
> >
> > Thank you for your response. Reading the other reviews, I agree that the full pseudo-code should appear, even if just in an appendix. A couple of questions: I am personally not very familiar with the theory of KDE-based (or regularised) particle filters, does consistency hold in the particle filtering case of recursive approximation? If such results exist they could be mentioned in the text. I missed the details on the adaptive MDPF on first reading. How does this strategy differ from the well known auxiliary particle filter [Pitt and Shepard 1999], that also uses a different distribution for weighting and resampling, other than that the distributions are both learned rather than analytically optimised?

---

> > > ### Author Response · Authors · 2024-08-12
> > >
> > > Thank you for taking the time to respond to our rebuttal.  We will add pseudo-code to the appendix in later revisions of this paper.
> > >  To answer your questions:
> > >
> > > **I am personally not very familiar with the theory of KDE-based (or regularised) particle filters, does consistency hold in the particle filtering case of recursive approximation?**
> > >
> > > We did not emphasize this due to space constraints, but the theoretical foundations of KDE-based particle filters (which are standardly called “regularized” particle filters in the literature) are quite strong.  The cited paper by Musso et al. [40] gives an overview, and detailed results are developed by Le Gland and Oudjane (Annals of Applied Prob. 2004).  For conventional discrete-resampling particle filters, theoretical results show weak convergence to the optimal filter posterior as the number of particles becomes large.  Regularized particle filters have similar guarantees, but by directly approximating the posterior density, one can also obtain strong convergence (in variational distance or other Lp norms) to the optimal posterior.  Moreover, while standard particle filters require strong “mixing” assumptions on the dynamics to avoid degeneracy, regularized filters behave better (in both theory and practice) for low-noise dynamics due to the diversity provided by their continuous resampling step.
> > >
> > > In this context, one can think of the KDE smoothing as adding a small amount of regularization/bias to the density estimate in order to reduce variance.  For classic regularized particle filters, this bias is decayed to zero as the number of particles increases by reducing the kernel bandwidth (theorems characterize the optimal convergence rate).  For our mixture density particle filters/smoothers, the bandwidth is implicitly tuned to the given application and particle budget via the training process.
> > >
> > > **I missed the details on the adaptive MDPF on first reading. How does this strategy differ from the well known auxiliary particle filter [Pitt and Shepard 1999], that also uses a different distribution for weighting and resampling, other than that the distributions are both learned rather than analytically optimised?**
> > >
> > > Great question.  The classic auxiliary particle filter is a generative model that tries to look “one step ahead” in the discrete resampling step, by approximating the likelihood of the next observation.  However to do this, it needs to be able to quickly compute a mean/mode of the dynamics, which is easy for many classic models (e.g., if dynamics are a fixed non-linear function plus Gaussian noise) but not for our scenario (where dynamics are parameterized by a learned neural network and may be complex, see Fig. 5).
> > >
> > > In contrast, our tuning of the KDE resampling bandwidth is not determined by a one-step lookahead of the next observation, but instead by the behavior of the particle filter over many future time steps (as quantified by backpropagation of loss gradients).  Interestingly, it is not clear how to do this tuning based on theoretical results for standard particle filters, so we are unaware of direct analogs of our approach in the classic particle filter literature.
> > >
> > > In some sense, our adaptive proposals and the auxiliary PF are complementary to each other, and in future research it would be interesting to explore variants of the auxiliary PF idea that accommodate the more complex dynamical models we consider.
> > >
> > > **Reference:**
> > >
> > > Le Gland, F., & Oudjane, N. (2004). Stability and uniform approximation of nonlinear filters using the Hilbert metric and application to particle filters. The Annals of Applied Probability, 14(1), 144-187.

---

### Official Review · Reviewer_tGtz · 2024-07-20

**Soundness:** 2
**Presentation:** 3
**Contribution:** 2
**Rating:** 3
**Confidence:** 4

**Summary:**

The paper proposes a learnable differentiable particle smoother system that extends an existing differentiable particle filter to a smoother. The method utilizes two independent particle filters for the two smoothing directions with importance sampling to address computation scaling issues.

**Strengths:**

- Proposed smoothing approach outperforms filtering setting
- Combination of sensible components into the proposed filtering setup

**Weaknesses:**

While the proposed method appears to work and improves over the baseline particle filter variant it is unclear how well it performs in contrast to other smoothers which would pose a fair comparison. Additionally, the particular task of large scale global localization is something that robotics has been investigating for a long time and as such one would expect to see comparisons to the typical way this is solved, such as full visual SLAM or hybrid place recognition with localization. As the KITTI dataset is used it is also disappointing that no comparison to the benchmark results of the dataset are presented. Overall it is unclear how well the proposed method works when compared to known approaches deployed in practice.

The power of a learnable filter/smoother is that it can learn models where it is hard or impossible to define them manually. However, the chosen application has very well defined and workable models making the motivation less convincing. As such it would be highly interesting to see the method be evaluated on scenarios where no models exist or even exist and are compared with. In line with that it would also be interesting to know how feasible it is to run the proposed method on higher dimensional data.

The method requires a significant amount of time to train and run which makes it questionable whether it is even practical for real applications. Another issue arises from this is that different methods were provided with different amount of training time. While it is valid to see how well methods work when given the same parameters (particles) it is also important to assess how good other methods would perform if they had access to the same amount of training time which for the MDPF vs MPDS is still twice as much.

The paper mentions that given true dynamics and likelihood models the sampling will compensate for the mismatch between samples and density mixture. Is that statement meant to indicate that these ground truth models are required, or that in an ideal case this would technically happen? If the first meaning is intended, where do these models come from? If the second meaning is intended, how close does the proposed method get to this and is there a way to assess that?

It appears that the absolute error of the proposed method and all others is exceedingly large with being many meters off to achieve any reasonable amount of recall. This is where reproducing numbers from the KITTI benchmark is crucial to get an idea of the hardness of the problem.

Figure 5 is hard to interpret, especially since there are now comparisons to other methods provided. While one can see that the prediction somewhat maps to parts of the future state it is very noisy and unclear how good it actually is.

Given the massive training times of the method, it would be important to provide insights into the scalability both in terms of number of particles and dimensionality of the estimation problem. Another question that arises in that regard is how the proposed method compares to methods based around Stein variational inference [1] which is capable of handling high-dimensional problems very efficiently.

Overall, while the proposed method performs better than other filtering methods it is unclear how practical the method is given the required compute time and limitation in the dimensionality of problems. This is further exarberated by the fact that there is no comparison to typical solutions of the chosen problem setups and a lack of particle smoother baselines.

[1] Liu, Qiang, and Dilin Wang. "Stein variational gradient descent: A general purpose bayesian inference algorithm." *Advances in neural information processing systems* 29 (2016).

**Questions:**

- With regards to the sampling, in the paper M is selected to be equal between the two filter directions. What's the impact of selecting M and it's distribution onto the two distributions?
- In the description of baseline methods it is mentioned that refinement methods require an "accurate initial estimate". How good does this have to be in practice and/or in the specific experiments shown in the paper?

**Limitations:**

The limitation section is very basic and really is about the limitation of the family of methods in general rather than the proposed method. Expanding on that and providing information about the proposed method would be expected.

---

> ### Author Rebuttal · Authors · 2024-08-06
>
> Thank you for your comments and feedback.  Please see our global response for discussion of training time and paper figure formatting.
>
> With regard to comparison with existing works, we would like to make the distinction between local localization and global localization.  Local localization estimates the state in relation to the starting point. Global localization requires estimating the state in relation to a global map origin. Visual SLAM systems almost exclusively solve the local localization task, using the starting position as the origin of their estimated map. If a map is provided, then just the localization part of Visual SLAM can be run, but detailed visual or 3D maps of the environment are needed.  These have prohibitive memory requirements at city scales and need constant updating as the visual appearance of the environment changes (i.e. with the weather/seasons) [11].  Hybrid place recognition with localization also requires detailed visual or 3D maps [Wang et al., 2023]. Instead we seek to use planimetric maps for global localization, which are compact and robust to environment changes.  It is not obvious how to apply SLAM/Hybrid place recognition systems to this type of map.
>
> We use the KITTI dataset in a way that makes comparison against the KITTI leaderboard impossible. The leaderboard ranks methods for local localization with no prior map provided. The methods seek to only accurately estimate relative position without regard to global position. In contrast, our method solves the global localization task using a map where the starting state itself is uncertain. Comparison of our method to the leaderboard would not be appropriate, as we are completing a different task. We instead compare against state-of-the-art prior methods for global localization (see Figs. 3 and 6) to illustrate task difficulty.  Compared to the prior state-of-the-art [11], we have both higher accuracy and lower localization cost at test-time.
>
> Figure 5 highlights the effectiveness of the learned complex, nonlinear dynamics model at propagating particles given noisy actions. Particles are spread to ensure some overlap with the true state, and the measurement model then concentrates probability mass over the true state using the latest observation.  Other methods do not have dynamics models and thus similar figures cannot be made.
>
> In appendix B, we show some scalability results for the number of particles used during inference. Interestingly, models can be trained with few particles to limit computation requirements during training, but then many particles can be used during inference to increase accuracy. In this work, we focus on algorithmic innovations over prior differentiable particle filters/smoothers, and apply our MDPS to 3D problems.  However, in future work we plan to explore scalability to higher-dimensional problems.
>
> Stein variational inference iteratively transforms particles from a starting distribution to samples from a known target distribution using the score function, and can be thought of as a Monte Carlo method that avoids computing normalizing constants. Although this method has been applied to state estimation [Maken et al. 2022, Fan et al. 2021], these methods assume the prior and dynamics models are fixed and known, and thus could not be applied to our discriminative training scenario where models must be learned from data.
>
> **To answer your questions:**
>
> - In section 4 we say *“Given true dynamics and likelihood models, importance sampling may correct for the fact that smoothed particles are drawn from a mixture rather than a product of filtered densities...”*. We are saying that we wish for samples from the product of densities (hard to sample from) but draw samples from a mixture of densities (easy to sample from) and use importance sampling [21] to compensate for this mismatch by weighting the drawn samples.  If the true dynamics and likelihood models are known then importance sampling would produce weighted samples from the true posterior distribution, but since these models are learned the weighted samples are from an approximation of the true posterior distribution. There is no easy way to assess the accuracy of the learned posteriors, since the true distributions are unknown.
>
> - See appendix B for experiments varying the number of particles $M$.  Sufficient number of particles are needed to accurately represent the forward and backward filter distributions.  $M$ that is too small can negatively impact MDPS by limiting the representational power of the two filters, and thus increasing variance when estimating gradients. Increasing $M$ can only help performance by increasing representation power.  We empirically set $M$ to be equal for both filters in order to simplify implementation, however tuning $M$ separately for each direction could potentially provide computational savings.
>
> - Refinement methods require an accurate initial estimate.  [11] sets this to be within a 64x64 meter box centered around the true position and within 45 degrees of the true bearing in order for the method to converge. In [9] this box is decreased to 40x40 meters, and the authors state that if the initial state is far from the true state, extracted features from the image and the map will be too dissimilar and the refinement method is often trapped in poor local optima. In contrast, we initialize our MDPS method with large uncertainty (particles randomly within 150 meters from the true state).
>
> **Additional References:**
>
> Y. Wang, Y. Qiu, P. Cheng and J. Zhang, "Hybrid CNN-Transformer Features for Visual Place Recognition," in IEEE Transactions on Circuits and Systems for Video Technology, 2023
>
> Maken, Fahira Afzal, Fabio Ramos, and Lionel Ott. "Stein particle filter for nonlinear, non-gaussian state estimation." IEEE Robotics and Automation Letters, 2022
>
> Fan, Jiaojiao, Amirhossein Taghvaei, and Yongxin Chen. "Stein particle filtering." arXiv preprint arXiv:2106.10568 (2021).

---

> > ### Comment · Reviewer_tGtz · 2024-08-13
> >
> > Thank you for the clarifications. While I understand the points I am currently not fully convinced by them. I can see that the proposed method works and could be useful, yet I do not truly see how the application used to showcase it is a good one. At this stage I will not change my score but will keep an open mind during the reviewer discussion.

---

### Author Rebuttal · Authors · 2024-08-06

Thank you all for your feedback and helpful suggestions.  We want to address a few topics that were referenced by multiple reviewers.

First, we would like to emphasize that our work is the first and only generalization of state-of-the-art differentiable particle filters to the more challenging particle smoothing scenario.  It is for this reason that most of our experimental comparisons are not to other particle smoothers, but to state-of-the-art methods for the global visual localization task, where we find substantial advantages in both speed and accuracy.

We agree that some formatting aspects of the paper can be improved. Unfortunately due to space constraints, the size of some figure elements and fonts was limited. In future revisions, we will be sure to make changes to the figures to increase the size of renderings and fonts for better readability (moving some details to appendices if necessary).

With regard to training time, we train all methods to convergence. During training, we monitor the training loss, and decay the learning rate each time a training loss plateau is detected. Training is terminated once the training loss plateaus and the learning rate is sufficiently small that decreasing it has no effect on training.  Finally, we use validation data to select the best model to use for evaluation (to prevent overfitting).  We find that given this training methodology, additional training time for the various methods would not yield better performance as all methods were trained to convergence (i.e. given as much training time as needed to produce the best model). The improved performance of MDPS over MDPF is not due to additional training time, but rather due to better dynamics and measurement models being learned as well as using the backward filter to propagate information from future observations (i.e., using the full sequence to estimate the state posterior distribution). Once trained, MDPF is very efficient during inference as shown in [14]. Since our MDPS simply runs two MDPFs and an additional combination step, it is also efficient during inference, making real-world use more practical than other methods such as dense search [11].

---

### Decision · Program_Chairs · 2024-09-25

**Decision:**

Accept (poster)

**Comment:**

This paper presents a novel end-to-end differentiable particle smoother called Mixture Density Particle Smoother (MDPS) for state estimation problems in robotics and computer vision. The key contributions are:

1. Extending recent advances in differentiable particle filters to the more complex particle smoothing scenario.
2. Developing a gradient propagation framework across long time sequences for training particle smoothers.
3. Integrating forward and backward particle streams with stratified resampling and importance weighting.
4. Demonstrating improvements over state-of-the-art particle filters and search-based baselines on real-world, city-scale global vehicle localization tasks, albeit under the assumption that the localization is over a planimetric global map.

The reviewers were split in their assessments, with two recommending acceptance and two recommending rejection. The positive reviews highlighted the novelty of the approach, the strong experimental results and specifically the performance gains over rfiltering-only approaches, and the potential impact on real-world applications. The negative reviews raised concerns about the clarity of the mathematical presentation and the technique/claims, and questions about the practical applicability given the computational demands.

The authors' rebuttals addressed many of these concerns, providing additional details on the mathematical foundations, clarifying the novelty of their approach, and explaining the experimental setup. They also acknowledged limitations and areas for future work. While there were specific concerns regarding insufficient baseline comparisons, the authors state their approach operates under significantly more challenging conditions (e.g. estimation over a much larger parameter counts). There has been a concern regarding the definition of stratified resampling, which despite author responses, does not seem to have been fully resolved.

Despite the split opinions, I recommend accepting this paper with the condition that the authors address the clarity and rigor concerns in a revised version. The reasons for this recommendation are:

1. This is the first work to successfully develop an end-to-end differentiable particle smoother, which is a non trivial contribution to the field, whereas differentiable particle filters do exist.
2. The method demonstrates substantial improvements over baselines on tasks that are moderately challenging.
3. The approach has the potential to address important problems in robotics and computer vision with potential real-world applications.
4. The authors have actively participated in the discussion to help alleviate reviewer concerns, and have shown willingness to address reviewer concerns and improve the paper.

To address the remaining concerns, the authors should:
1. Improve the clarity and rigor of the mathematical presentation, particularly around gradient estimations, and the definition of stratified resampling in the context of the this paper.
2. Add more detailed comparisons to classical particle smoothing methods, even if only in synthetic scenarios.
3. Provide a more in-depth discussion of the computational requirements and potential limitations in high-dimensional settings, while also highlighting the fact that inference is more efficient.
4. Consider adding pseudocode for the algorithm to improve reproducibility.

In conclusion, while there are areas for improvement, the novel contributions and strong empirical results make this paper a valuable addition to the NeurIPS proceedings. The authors should use the revision period to address the noted concerns and strengthen the paper further.